# Histone H3.3 deposition in seed is essential for the post-embryonic developmental competence in *Arabidopsis*

Ting Zhao[1,4], Jingyun Lu[1,4], Huairen Zhang[1,4], Mande Xue[1,2], Jie Pan[1,2], Lijun Ma[1,2], Frédéric Berger ®[3] & Danhua Jiang ®[1,2] ✉

The acquisition of germination and post-embryonic developmental ability during seed maturation is vital for seed vigor, an important trait for plant propagation and crop production. How seed vigor is established in seeds is still poorly understood. Here, we report the crucial function of *Arabidopsis* histone variant H3.3 in endowing seeds with post-embryonic developmental potentials. H3.3 is not essential for seed formation, but loss of H3.3 results in severely impaired germination and post-embryonic development. H3.3 exhibits a seed-specific 5' gene end distribution and facilitates chromatin opening at regulatory regions in seeds. During germination, H3.3 is essential for proper gene transcriptional regulation. Moreover, H3.3 is constantly loaded at the 3' gene end, correlating with gene body DNA methylation and the restriction of chromatin accessibility and cryptic transcription at this region. Our results suggest a fundamental role of H3.3 in initiating chromatin accessibility at regulatory regions in seed and licensing the embryonic to post-embryonic transition.

The formation of seeds is one of the most important innovations that contribute to the success of flowering plants. By seed formation, the embryo is protected in a desiccated and quiescent state, and, meanwhile, ready to swiftly initiate germination and post-embryonic development when optimal growth condition arises[1]. Producing highly vigorous seeds is thus key for plant reproduction and agriculture. Embryo obtains its germination capacity during the late seed maturation stage[2]. However, the underlying mechanisms establishing germination and post-embryonic developmental competence in seeds remain poorly understood.

The eukaryotic genome is tightly packed in the form of chromatin, which consists of DNA and histone-formed nucleosomes as basic subunits. Chromatin compaction restricts the access of transcription factors to *cis*-regulatory elements such as promoters and enhancers, and thus affecting gene transcription[3,4]. During cellular differentiation and developmental transitions, epigenetic reprogramming induces chromatin structure changes, thereby modifying the accessibility of regulatory elements to transcription machinery[5,6]. The establishment of chromatin accessibility is thus a pioneer event in developmental transition and lineage-specific transcriptional regulation. Several mechanisms have been identified to control chromatin structure dynamics, including ATP-dependent chromatin remodeling, post-translational modifications on histones, and the exchange of histone variants[7,8]. Besides their sequence divergence, histone variants often possess different expression patterns and in some cases unique modifications[9–11]. The deposition of histones variants can modify nucleosome stability, which differentiates chromatin into specialized property and cellular function[4,12].

In animals, the histone H3 variant H3.3 differs from the replicative H3.1/H3.2 with only 4–5 amino acids[13]. Unlike H3.1, which is DNA

[1]State Key Laboratory of Plant Genomics, Institute of Genetics and Developmental Biology, The Innovative Academy for Seed Design, Chinese Academy of Sciences, Beijing, China. [2]University of Chinese Academy of Sciences, Beijing, China. [3]Gregor Mendel Institute, Austrian Academy of Sciences, Vienna BioCenter, Dr. Bohr-Gasse 3, 1030 Vienna, Austria. [4]These authors contributed equally: Ting Zhao, Jingyun Lu, Huairen Zhang. ✉e-mail: dhjiang@genetics.ac.cn

replication-dependently expressed and incorporated into the chromatin, H3.3 is expressed and incorporated in a DNA replication-independent manner[14–18]. H3.3 is associated with transcribed regions and gene regulatory elements[19,20]. In *Xenopus*, H3.3 is essential for gastrulation, and it is proposed that H3.3 creates a permissive chromatin state critical for embryo development[21,22]. Knockout one of the two H3.3-coding genes in mouse causes reduced viability and infertility[23–25], and complete deletion of H3.3 results in early embryonic lethality[26]. H3.3 is required to maintain a decondensed chromatin state and active histone modifications in mouse embryo[27]. Nevertheless, loss of H3.3 seems to have a limited impact on gene transcription, rather it causes abnormal chromosome segregation and impaired genome integrity[26,27].

Although evolved independently in separate kingdoms, H3.1 and H3.3 in higher plants acquired comparable features as their counterparts in animals through convergent evolution[28–30]. For instance, in *Arabidopsis* H3.3 differs from H3.1 with four amino acids[31], and H3.3 and H3.1 are DNA replication-independently and dependently expressed respectively[32]. Moreover, *Arabidopsis* H3.3 is mostly enriched at the body of transcribed genes[32,33]. Three genes *HTR4*, *HTR5* and *HTR8* in *Arabidopsis* encode H3.3. Knockdown of *Arabidopsis* H3.3 (*h3.3kd*) results in mild developmental defects, including leaf serration, early flowering and slightly reduced fertility[34,35]. H3.3 promotes permissive histone modifications at the floral repressor *FLOWERING LOCUS C* (*FLC*), leading to its activation to repress flowering[35]. Compared with the broad distribution of H3.3 on transcribed genes, only hundreds of genes are misexpressed in *h3.3kd*, indicating that H3.3 may not regulate transcription directly[32]. Hence, the role of H3.3 in plants remains to be investigated.

Here, we report the critical function of *Arabidopsis* H3.3 in the establishment of chromatin accessibility and endowing seeds with post-embryonic developmental competence. Complete deletion of H3.3 shows no obvious impact on embryogenesis, but results in severe defects in germination and vegetative growth. H3.3 is enriched around both 5′ and 3′ gene ends in mature seeds. The 5′ enrichment of H3.3 is seed-specific, and it facilitates chromatin opening, particularly at the regulatory regions of responsive and post-embryonic developmental genes, and H3.3 is essential for their transcriptional regulation during germination. On the other hand, 3′ localization of H3.3 represents a canonical H3.3 deposition pattern that correlates with gene body DNA methylation and the restriction of chromatin accessibility and cryptic transcription. Our results support a fundamental role of H3.3 in the establishment of chromatin accessibility and gene transcriptional regulation important for post-embryonic development.

## Results

### H3.3 is necessary for plant post-embryonic development

In a previous study, we created a complete H3.3 knockout mutant by combining a CRISPR-Cas9 generated *htr4;htr5* double mutant with an *htr8* mutant carrying a T-DNA insertion at *HTR8*. At that point, we did not obtain triple homozygous plants when screening the *htr4/ htr4;htr5/htr5;htr8/+* (*h3.3ko/+*) progenies, and thus we speculated that the *h3.3ko* mutant was embryonic lethal[34]. To analyze *h3.3ko* mutant phenotypes in detail, we examined the embryo development on *h3.3ko/+* plants. Surprisingly, all seeds developed without abortion (Supplementary Fig. 1a). Around 7.5% to 11.5% of seeds collected from *h3.3ko/+* plants could not germinate after 7 days of imbibition, and thus these were likely *h3.3ko* (Fig. 1a) (Supplementary Fig. 1b). This ratio is lower than 25% because knockout of H3.3 partially impairs male gametogenesis, and when crossing *h3.3ko/+* with the wild-type (WT) Columbia (Col) mother, the transmission of *h3.3ko* was reduced to around 16% instead of the expected 50%[34]. These ungerminated seeds were then subjected to prolonged imbibition (up to two months), a portion of these seeds eventually germinated and produced small seedlings, and genotyping analysis confirmed that these seedlings

were *h3.3ko* (Fig. 1b, c). Most of the *h3.3ko* seedlings stopped development immediately after germination without expanding cotyledons, and only a few of them could produce two true leaves (Fig. 1c, d). In rare cases, *h3.3ko* plants developed until the bolting stage and produced an inflorescence-like structure (Fig. 1c and Supplementary Fig. 1c). However, these plants ceased growth afterward and no silique was generated.

In order to efficiently obtain the *h3.3ko* mutant seeds, we transformed a construct carrying a WT *pHTR5::HTR5* and a mature seed stage-specific mCherry fluorescence selection marker into *h3.3ko/+*[36]. The WT *HTR5* successfully rescued the *h3.3ko* phenotypes (Fig. 1d, e). *Arabidopsis* H3.3 differs from H3.1 with only four amino acids. To examine whether the *h3.3ko* phenotypes were specifically caused by the loss of H3.3 or merely due to the reduced H3 supply, we attempted the *h3.3ko* complementation by expressing *HTR13* (an H3.1-coding gene) under the *HTR5* promoter (Supplementary Fig. 1d). Expressing *HTR13* only slightly accelerated the germination of *h3.3ko* (Fig. 1e), suggesting that the *h3.3ko* defects were largely H3.3 specific. We then expressed *HTR5* using the *HTR13* promoter, which was successfully employed in driving *HTR13* expression to complement the *h3.1kd* phenotypes[37]. Nevertheless, expressing *HTR5* under the *HTR13* promoter also failed to rescue the *h3.3ko* germination defects (Fig. 1e, Supplementary Figs. 1d and S1e). Together, we conclude that both the H3.3 protein sequences and its expression pattern are required for its function in seed germination.

We collected seeds from heterozygous *pHTR5::HTR5* transgenic plants in the *h3.3ko* background, from which the *h3.3ko* seeds could be identified by selecting mature seeds without the mCherry signal (Supplementary Fig. 1f). The *h3.3ko* seeds selected in this way behaved the same as those segregated from *h3.3ko/+* (Fig. 1a, b, d and e), and hence were used in all subsequent studies.

All germination tests were performed with seeds stored for three months after harvesting (after-ripened). Moreover, seeds were stratified for two days during germination, further excluding the possibility that the *h3.3ko* germination defects were due to its strong dormancy. Endosperm and testa (seed coat) are critical tissues in dormancy maintenance, and they impose a physical constraint on the radicle protrusion[38,39]. To examine which tissues are responsible for the *h3.3ko* germination defects, we isolated *h3.3ko* embryos by removing endosperm and testa. The isolated *h3.3ko* embryos showed much slower growth compared with WT (Fig. 1f), indicating strong defects in *h3.3ko* embryos.

### *h3.3ko* shows strong defects in transcriptome

To search for a mechanism to explain the strongly impaired germination in *h3.3ko*, we profiled transcriptomes of WT and *h3.3ko* using RNA-seq. Seeds stored for three months after harvesting (hereinafter referred to as mature seeds) were subjected to imbibition and RNA was extracted from seeds collected at various time points (Fig. 2a). Principle component analysis (PCA) showed that compared with WT, the overall transcriptome changes in *h3.3ko* were less dynamic during imbibition (Fig. 2b). We combined all transcript level significantly decreased and increased genes in *h3.3ko* (fold change >2 and *P* adjust <0.05) identified from each time point during imbibition and performed gene ontology (GO) analysis, and found that responsive genes were especially enriched (Fig. 2c) (Supplementary Data 1), suggesting compromised imbibition response in *h3.3ko* compared with WT. GA and ABA are two major hormones that antagonistically regulate germination, during which GA biosynthesis and ABA catabolic genes need to be activated[40,41]. However, the activation of key GA synthetic and ABA catabolic genes such as *GA3OX1*, *GA3OX2*, *GA20OX1*, *GA20OX2*, and *CYP707A2* was strongly abolished in *h3.3ko* (Fig. 2d and Supplementary Fig. 2a), in line with its impaired germination. Intriguingly, the variation between *h3.3ko* and Col transcriptomes was already notable at the mature seed stage and nearly 5000 genes were significantly

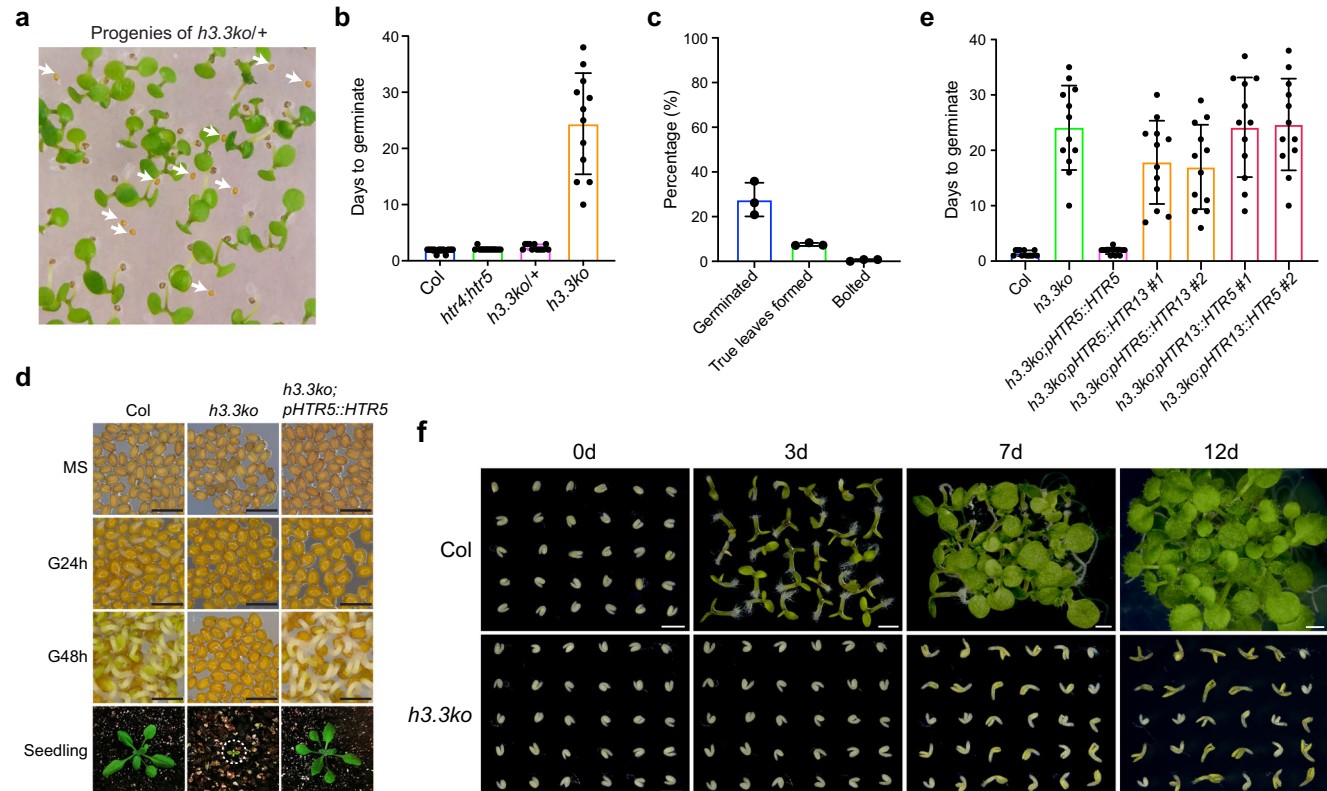

**Fig. 1 | H3.3 is required for post-embryonic development. a** The germination phenotype of *h3.3ko/+* (*htr4/htr4;htr5/htr5;htr8/+*) progenies. Arrows indicate *h3.3ko* seeds that are strongly delayed in germination. **b** Days to germinate of Col, *htr4/htr4;htr5/htr5*, *h3.3ko/+* (*htr4/htr4;htr5/htr5;htr8/+*) and *h3.3ko* (*htr4/htr4;htr5/htr5;htr8/htr8*) seeds. Seeds collected from *h3.3ko/+* plants were sown on 1/2 MS, their genotypes were determined with seedlings formed after germination, and the first 12 germinated seeds were scored for each genotype. Values are means ± SD. (*n* = 12 biologically independent seeds). Radicle protrusion was considered as the completion of seed germination. **c** Percentages of *h3.3ko* seeds that germinated (within two months), produced true leaves and bolted. Values are means ± SD of three biological replicates. At least 110 seeds were used for each replicate. **d** The

germination and growth phenotypes of Col, *h3.3ko* (segregated from *h3.3ko;pHTR5::HTR5/-*) and *h3.3ko;pHTR5::HTR5*. Scale bars = 1 mm (for seeds) or 1 cm (for seedlings). MS: mature seeds, G24h: subjected to germination for 24 h, G48h: subjected to germination for 48 h. **e** Days to germinate of Col, *h3.3ko* (segregated from *h3.3ko;pHTR5::HTR5/-*) and transgenic seeds expressing *HTR5* or *HTR13*. The first 12 germinated seeds were scored. More than 110 seeds were sown on 1/2 MS. Values are means ± SD. (*n* = 12 biologically independent seeds). Radicle protrusion was considered as the completion of seed germination. For *h3.3ko;pHTR5::HTR13* and *h3.3ko;pHTR13::HTR5*, the results from two independent transgenic lines are presented. **f** The germination and growth phenotypes of isolated Col and *h3.3ko* embryos. Scale bars = 1 mm. d: days.

---

misexpressed in mature *h3.3ko* seeds (Fig. 2b, e). Therefore, H3.3 is already required for the establishment of proper transcriptome in mature seeds.

## A distinctive pattern of H3.3 enrichment over genes in mature seeds

To test whether the impaired germination of *h3.3ko* is due to defects in its mature seeds or is caused by the lack of H3.3 during imbibition, we introduced into *h3.3ko/+* an *HTR5* expression construct based on an estradiol inducible system (*pER8::HTR5*), which induces a strong uniform gene expression pattern similar to that of *HTR5*[32,42,43]. Seeds collected from *h3.3ko/+;pER8::HTR5* were subjected to imbibition with or without estrogen, and *HTR5* transcription was successfully induced by estrogen (Fig. 3a). Interestingly, the induction of *HTR5* expression during imbibition showed no impact on the germination of *h3.3ko* (Fig. 3b, c). The transcript levels of genes involved in H3.3 deposition were comparable in WT and *h3.3ko*, indicating that the H3.3 deposition pathway is generally intact in *h3.3ko* (Supplementary Fig. 2b). Therefore, it appears that loss of H3.3 already caused strong defects in mature seeds that affect germination, and a supply of H3.3 to *h3.3ko* during imbibition is not sufficient to complement the defects caused by the loss of H3.3 in seeds.

The above results prompted us to examine if there were any morphological defects in *h3.3ko* seeds. The seed size and seed weight

of mature *h3.3ko* seeds were comparable to that of WT (Supplementary Fig. 3a–c). In addition, mature embryos isolated from WT and *h3.3ko* seeds were morphologically similar (Fig. 1f and Supplementary Fig. 3d). Seed storage proteins (SSPs) are synthesized during the late stages of seed development and are the major nutritional sources for subsequent germination and early seedling development[44]. The abundance of SSPs was examined with mature seeds, and the major SSPs including 12S globulins and 2S albumins were normally accumulated in *h3.3ko* (Supplementary Fig. 3e), suggesting that H3.3 is not required for their synthesis. Hence, loss of H3.3 did not affect embryo morphology and the accumulation of SSPs.

In WT, the transcript levels of H3.3-coding genes markedly increased towards the end of embryo development, contrasting with the decreased expression of H3.1-coding genes (Fig. 3d, Supplementary Fig. 1d, e)[45]. Since H3.3 is not required for seed formation (Supplementary Figs. 1a and 3), we reasoned that H3.3 could become critical at the late seed maturation stage soon before seeds turn fully mature to prepare them for germination. To test this possibility, we expressed H3.3 in *h3.3ko* with the promoter of *at2S3* (a seed storage protein gene) or *H2B.S* (a seed-specific H2B variant gene), both of which are expressed specifically at the late stages of seed development[44,46]. Expressing *HTR5* under either promoter increased the germination rate of *h3.3ko*, with the *H2B.S* promoter showing stronger capability (Fig. 3e, f and Supplementary Fig. 4a). However, in

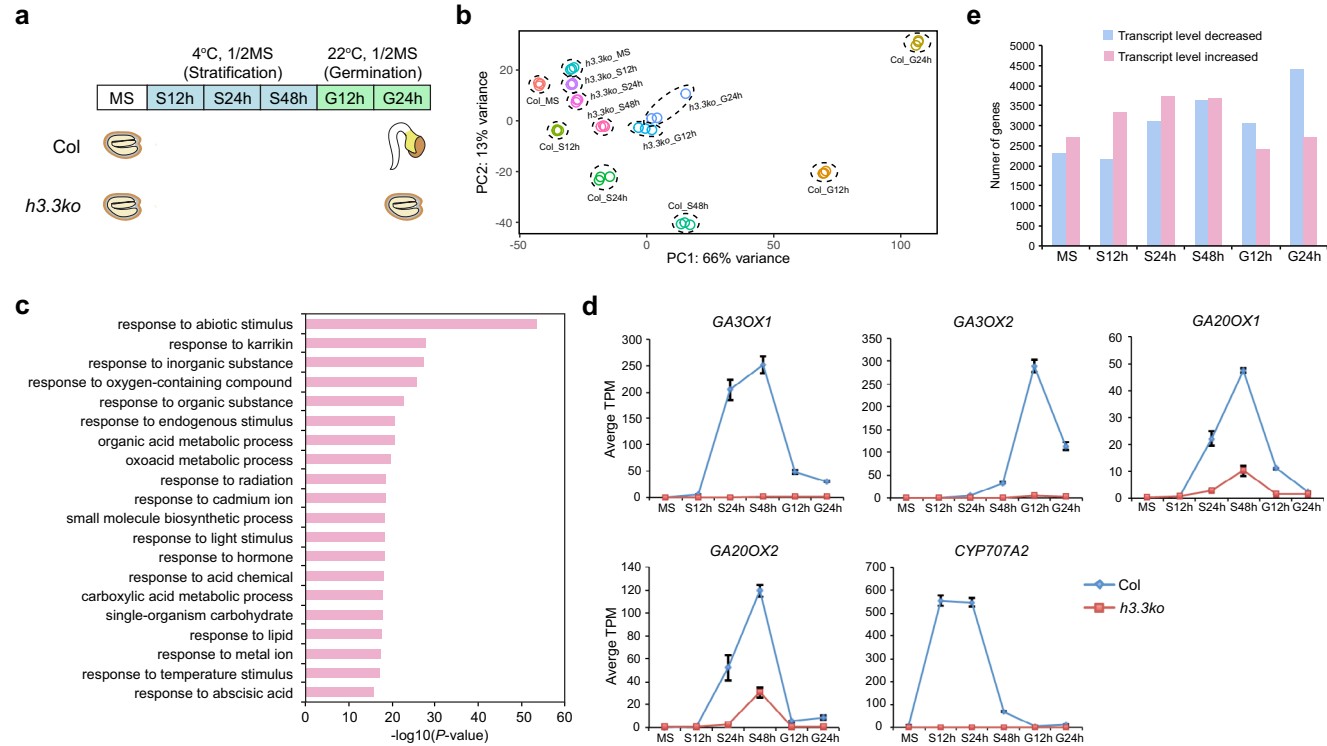

**Fig. 2 | Transcriptome changes in *h3.3ko*. a** The germination time course in this study. MS: mature seeds, S12h/S24h/S48h: stratified for 12/24/48 h, G12h/G24h: subjected to germination for 12/24 h. At G24h, the germination of Col but not *h3.3ko* has been completed. **b** PCA plot of Col and *h3.3ko* seeds imbibed for different times showing their transcriptome differences determined by RNA-seq. PC1 covers the highest amount of variance between samples; PC2 covers most of the remaining variance. Three biological replicates were performed for each line at each time point. **c** GO analysis of a combined set of all significantly misexpressed genes in *h3.3ko* identified from each imbibition time point. Top 20 representative terms are listed and ranked by *P* value, which was calculated with hypergeometric test. **d** Transcript levels of *GA3OX1*, *GA3OX2*, *GA20OX1*, *GA20OX2* and *CYP707A2* during imbibition determined by RNA-seq (TPM) in Col and *h3.3ko*. Values are means ± SD of three biological replicates. **e** Number of transcript levels significantly decreased and increased genes in *h3.3ko* compared with Col at different imbibition time points.

both cases, most of the seedlings ceased development after germination, similar to the *h3.3ko* seedlings. This is probably because the seedling development also requires H3.3 expressed during and after germination. Nevertheless, these results further support the notion that H3.3 expressed during seed maturation is necessary for seed germination.

Ideally, a comparison of WT and *h3.3ko* at the molecular level (e.g. RNA-seq) during seed maturation would help the identification of the earliest stage when *h3.3ko* starts to show defects. However, the *h3.3ko* seeds were morphologically identical to WT (Supplementary Figs. 1a and 3), making them impracticable to isolate until they were fully mature (Supplementary Fig. 1f). To this end, we focused on the mature seed stage, and profiled the genome-wide accumulation of H3.3 with an HTR5-GFP reporter line using ChIP-seq[32]. H3.3 was localized at euchromatin regions and depleted over transposable elements (TEs) in mature seeds (Supplementary Fig. 4b, c), a pattern similar to that in vegetative tissues[32,33,35]. Metagene analyses showed enrichment of H3.3 at both 5′ gene ends/promoters and 3′ gene ends (Fig. 3g, h and Supplementary Fig. 4d, e). Notably, the strong accumulation of H3.3 around the 5′ gene ends was not detected previously in vegetative tissues[32,33,35]. We thus further analyzed H3.3 enrichment during imbibition and in 10-day-old seedlings and observed a change in H3.3 enrichment patterns. In particular, H3.3 accumulation was gradually reduced at the 5′ gene end but increased at the gene body and the 3′ gene end (Fig. 3g, h and Supplementary Fig. 4d, e). These results suggest that H3.3 supplied during imbibition could be predominantly deposited to the gene body and the 3′ gene end but not the 5′ gene end, and probably explain why induction of *HTR5* expression during imbibition was not sufficient to rescue the germination defects of

*h3.3ko* (Fig. 3a–c). Together, we conclude that H3.3 enrichment around the 5′ gene end is specific to seeds, suggesting its importance at this stage of development.

## H3.3 regulates chromatin accessibility in mature seeds

The enrichment of H3.3 at the 5′ gene ends coincides with the localization of chromatin accessible regions around the gene transcription start sites[47,48], we thus examined chromatin accessibility changes in *h3.3ko* mature seeds by Assay for Transposase Accessible Chromatin sequencing (ATAC-seq). Overall, accessible regions were mainly enriched at euchromatin in both WT and *h3.3ko* (Fig. 4a). Chromatin accessibility over TEs was not changed in *h3.3ko* (Supplementary Fig. 5a), in line with the low H3.3 abundance at TEs. At genes, the majority of the open chromatin regions localized at 5′ gene ends/promoters in mature WT seeds (Fig. 4b–d and Supplementary Fig. 5b, c), similar to findings in vegetative tissues[47,48]. However, compared with WT, the chromatin accessibility was overall lower around the 5′ gene ends in *h3.3ko* and higher towards the 3′ ends (Fig. 4b–e and Supplementary Fig. 5b, c). Hence, loss of H3.3 results in a global change of chromatin accessibility at genes in mature seeds.

We further identified regions that had significantly altered chromatin accessibility in *h3.3ko* (fold change >2 and *P* adjust <0.05). 3278 and 3436 regions significantly lost or gained accessibility respectively (Supplementary Fig. 6a, b) (Supplementary Data 2). Less accessible regions in *h3.3ko* were preferentially localized at promoters, but regions with increased chromatin accessibility were mainly enriched at genic regions and 3′ gene ends (Supplementary Fig. 6c), parallel with the chromatin accessibility changes in *h3.3ko* over genes (Fig. 4b–e). Accessibility increased and decreased regions in *h3.3ko* were both

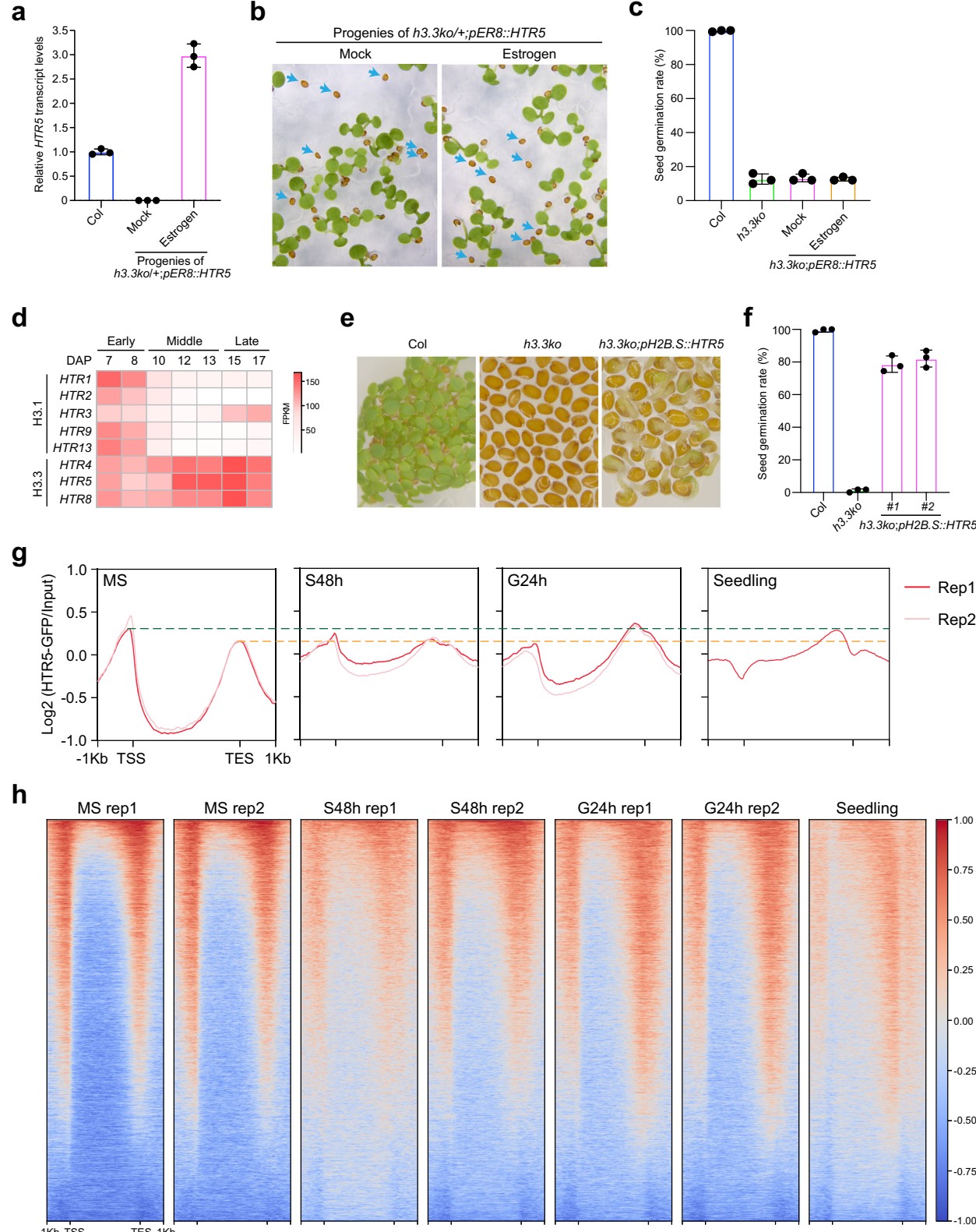

enriched with H3.3 in WT (Fig. 4e–g and Supplementary Fig. 6d, e), supporting the notion that H3.3 directly or indirectly regulates chromatin accessibility at these regions.

Notably, 5′ and 3′ localized H3.3 seems to have distinct impacts on chromatin accessibility. In animals, the coexistence of H3.3 and another histone variant H2A.Z destabilizes nucleosomes[49]. We thus profiled the enrichment of H2A.Z in mature seeds. Like in vegetative tissues, H2A.Z in mature seeds is enriched at the 5′ gene ends but depleted from the 3′ gene ends (Supplementary Fig. 7a)[50,51]. Moreover, accessibility decreased regions in *h3.3ko* carried higher levels of H2A.Z than accessibility increased regions (Supplementary Fig. 7b, c). To assess the requirement of H2A.Z in chromatin accessibility control, we performed ATAC-seq using mature seeds of WT and a hypomorphic *h2a.z* mutant, in which *HTA9*, one of the three H2A.Z-coding genes in

**Fig. 3 | H3.3 in seeds is essential for germination. a** The induction of *HTR5* by estrogen during imbibition. Seeds were imbibed with or without estrogen for 48 h. Values are means ± SD of three biological replicates. *PP2A* was used as an endogenous control for normalization. **b** The germination phenotype of *h3.3ko/ +;pER8::HTR5* progenies treated with or without estrogen during imbibition. Arrows indicate *h3.3ko;pER8::HTR5* seeds that are strongly delayed in germination. **c** Percentages of *h3.3ko;pER8::HTR5* seeds that germinated within 2 month of imbibition with or without estrogen. Values are means ± SD of three biological replicates. 50 seeds were assessed for each replicate. **d** Heatmap showing the transcript levels of H3.1 and H3.3-coding genes in dissected embryos at different (early/middle/late) developmental stages. DAP: day after pollination. The expression data were extracted from the published RNA-seq datasets[45]. **e** The germination phenotypes of Col, *h3.3ko*, and *h3.3ko;pH2B.S::HTR5* after subjected to germination for 84 h. **f** Percentages of *h3.3ko;pH2B.S::HTR5* seeds that germinated within 10 days of imbibition. Values are means ± SD of three biological replicates. At least 38 seeds were assessed for each replicate. **g** Metaplot of HTR5-GFP ChIP-seq signals over all genes in mature seeds, during imbibition, and in the 10-day-old seedling. TSS: transcription start sites, TES: transcription end sites. **h** Heatmap of HTR5-GFP ChIP-seq signals over all genes in mature seeds, during imbibition, and in the 10-day-old seedling.

---

*Arabidopsis*, is still expressed at low levels[50,52]. In *h2a.z*, a slight reduction of chromatin accessibility was observed at the 5′ gene end, while chromatin accessibility at the 3′ gene end was not changed compared with WT (Supplementary Fig. 8a). Importantly, regions with reduced accessibility in *h3.3ko* partially lost accessibility in *h2a.z* (Supplementary Fig. 8b), but loss of H2A.Z did not induce the chromatin accessibility at regions with increased accessibility in *h3.3ko* (Supplementary Fig. 8c). In addition, we observed moderately delayed germination of *h2a.z* compared with WT (Supplementary Fig. 8d). These results indicate that H2A.Z probably works together with H3.3 at the 5′ gene end to facilitate chromatin opening in mature seeds, while without H2A.Z at the 3′ gene end H3.3 somehow restricts chromatin accessibility. It is of note that the chromatin accessibility changes in the *h2a.z* mutant were less pronounced compared with that in *h3.3ko*. This could be because there is still residual H2A.Z in this *h2a.z* mutant, or other factors such as histone modifications enriched specifically at the 5′ or 3′ gene end may also contribute to the accessibility control together with H3.3.

**Altered chromatin accessibility in *h3.3ko* correlates with changes in transcription and gene body DNA methylation**

As mentioned above, the mature seed stage is the earliest time point when we could distinguish *h3.3ko* from WT (Supplementary Figs. 1a, f and 3). At this stage, the transcriptome of *h3.3ko* was already drastically different from that of WT (Fig. 2b, e). We then analyzed the correlations between changes in chromatin accessibility and transcriptome in mature *h3.3ko* seeds. In general, the distribution patterns of transcripts on genes were slightly altered in *h3.3ko* compared with WT, that the transcript levels were reduced around the 5′ gene end, and were increased towards the 3′ gene end (Fig. 5a, b, Supplementary Fig. 9a, b).

Next, we examined the transcripts of genes with significantly changed accessibility in *h3.3ko*. 4043 and 4388 genes were associated with accessibility decreased and increased regions respectively; with 920 genes associating with both (Supplementary Fig. 9c) (Supplementary Data 3). In *h3.3ko*, genes with accessibility decreased regions showed a strong reduction in chromatin accessibility around the 5′ end, and genes with accessibility increased regions mainly displayed a gain of accessibility towards the 3′ end (Supplementary Fig. 9d, e). These changes are consistent with the localization of accessibility decreased and increased regions in *h3.3ko* (Supplementary Fig. 6c). Overall, around accessibility decreased regions, fewer transcripts were detected in *h3.3ko* compared with WT (Fig. 5c), and genes associated with accessibility decreased regions showed an overall reduction in transcript levels (Fig. 5d). A detailed analysis revealed that a significant amount of them were indeed transcript level strongly decreased genes in mature *h3.3ko* seeds (Figs. 2e and 5e). Notably, some accessibility decreased region-associated genes showed increased transcript levels in *h3.3ko* compared with WT (Figs. 2e and 5e). Chromatin accessibility has also been found to link with gene repression[53,54], which may involve chromatin binding proteins such as transcription factors acting as repressors[55,56]. 2928 accessibility decreased region-associated genes did not show strong transcript level changes in mature *h3.3ko* seeds

compared with WT (Fig. 5e). Hence, these genes may bear regulatory capacity mainly necessary in later developmental stages.

At accessibility increased regions, transcript levels were overall higher in *h3.3ko* (Fig. 5c), and accessibility increased region-associated genes had more transcripts (Fig. 5d). Nevertheless, these transcripts produced in *h3.3ko* were unevenly distributed, with more being generated towards the 3′ gene end (Fig. 5d), indicating cryptic transcription. If this is the case, RNA Polymerase II (RNA Pol II) may randomly transcribe from both sense and antisense directions within these regions. We thus performed strand-specific RNA-seq to measure antisense transcripts in WT and *h3.3ko* mature seeds. The overall levels of antisense transcripts on genes were not increased in *h3.3ko* compared with WT (Supplementary Fig. 10a). However, at accessibility increased regions in *h3.3ko*, antisense transcripts were increased as well as sense transcripts (Supplementary Fig. 10b). Similarly, accessibility increased region-associated genes showed increased transcript levels from both sense and antisense, especially towards the 3′ gene end (Supplementary Fig. 10c). In mature seeds, RNA Pol II may remain at gene loci or still transcribe at a low rate[57]. We further selected a few genes showing increased chromatin accessibility and transcript levels towards the 3′ gene end and measured RNA Pol II enrichment levels at their loci by ChIP-qPCR in mature seeds (Fig. 5f). The results showed that the RNA Pol II enrichment levels were increased around the 3′ gene ends (Fig. 5g). Hence, the induction of transcripts at the 3′ gene end in *h3.3ko* is likely due to spurious transcription at both sense and antisense directions by RNA Pol II.

Knockdown of H3.3 induces the loss of CG DNA methylation at the gene body, especially towards the 3′ gene end[34]. Using genome-wide bisulfite sequencing (BS-seq), we confirmed a strong reduction of gene body CG methylation in mature *h3.3ko* seeds. In addition, CHG and CHH methylation levels at the gene body were also slightly reduced (Supplementary Fig. 11a). At TE regions, only a reduction of CHH methylation levels was observed in *h3.3ko* (Supplementary Fig. 11b). H3.3 is largely absent from TEs, and therefore the loss of CHH methylation at TEs could be caused by an indirect effect. Accessibility increased but not decreased regions in *h3.3ko* showed reduced CG methylation levels (Fig. 5g, h). Consistently, hypo CG differentially methylated regions (DMR) in *h3.3ko* displayed increased chromatin accessibility (Supplementary Fig. 11c). It is unlikely that loss of gene body CG methylation in *h3.3ko* causes increased chromatin accessibility since a complete loss of CG methylation in *met1* fails to induce chromatin accessibility at the gene body[58]. Together, these results suggest that H3.3-mediated chromatin accessibility control is probably required for proper transcriptional regulation and gene body DNA methylation.

**Loss of chromatin accessibility in *h3.3ko* is associated with gene misregulation during germination**

The above results indicate that H3.3 in seeds regulates chromatin accessibility and impacts on transcription. The enrichment of H3.3 around the 5′ gene end was gradually diminished along germination and vegetative growth (Fig. 3f, g). We thus examined chromatin accessibility in WT during germination and in seedlings, and found that the overall chromatin accessibility at the 5′ gene end was not strongly

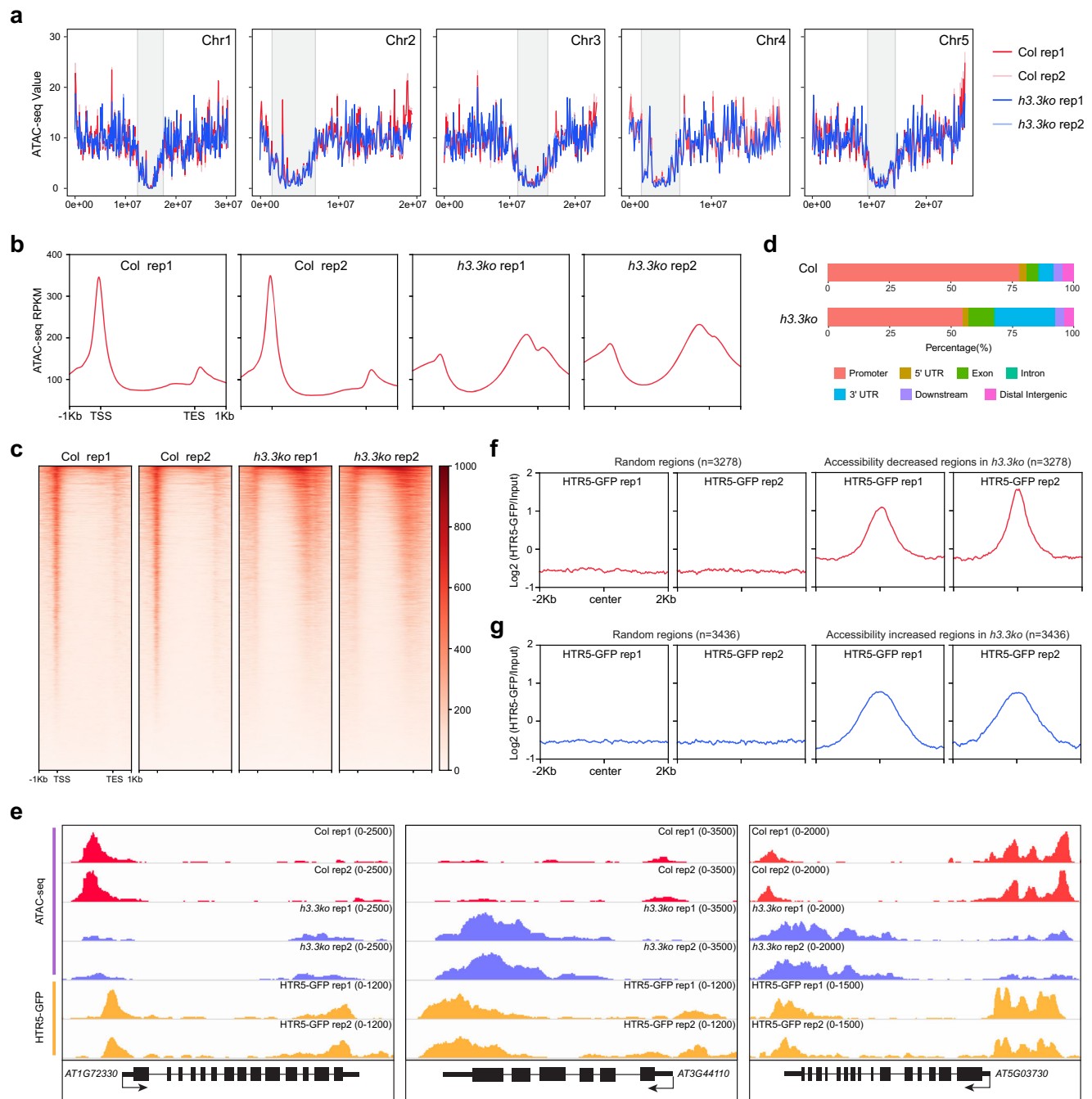

**Fig. 4 | Chromatin accessibility changes in mature *h3.3ko* seeds. a** ATAC-seq signals in Col and *h3.3ko* mature seeds over *Arabidopsis* chromosomes. Signals were calculated in 100 kb bins. Pericentromeric heterochromatin regions are indicated with gray shading. **b** Metaplot of ATAC-seq signals in Col and *h3.3ko* mature seeds over all genes. **c** Heatmap of ATAC-seq signals in Col and *h3.3ko* mature seeds over all genes. **d** Genomic distribution of accessible regions in Col and *h3.3ko* mature seeds. **e** Genome browser view of ATAC-seq and HTR5-GFP ChIP-seq signals in Col and *h3.3ko*

mature seeds over a few representative loci. *AT1G72330*: a gene mainly lost accessibility at the 5′ end/promoter region in *h3.3ko*, *AT3G44110*: a gene mainly gained accessibility around the 3′ end/genic region in *h3.3ko*, *AT5G03730*: a gene with both accessibility increased and decreased regions in *h3.3ko*. **f, g** Metaplot of HTR5-GFP ChIP-seq signals in mature seeds over accessibility significantly decreased (**f**) and increased (**g**) regions in mature *h3.3ko* seeds compared with Col. HTR5-GFP ChIP-seq signals over randomly selected regions are served as control.

changed (Supplementary Fig. 12a). We then focused on the H3.3-dependent open chromatin regions in mature seeds (accessibility significantly decreased regions in mature *h3.3ko* seeds that preferentially enriched around the 5′ gene end). At these regions, H3.3 enrichment levels were drastically reduced during germination and in seedlings, yet this was only coupled with a moderate reduction in chromatin accessibility (Fig. 6a, b). In *h3.3ko*, the accessibility at these regions remained low after imbibition (Fig. 6c and Supplementary Fig. 12b). These results indicate that the H3.3-dependent chromatin

accessibility in seeds is largely maintained during germination and subsequent development.

In mature seeds, many genes associated with the H3.3-dependent open chromatin regions did not show strong transcript level changes in *h3.3ko* compared with WT (Fig. 5e). To understand the importance of these H3.3-dependent open chromatin regions in transcriptional regulation during germination, we compared the transcript levels of their associated genes in WT and *h3.3ko* during imbibition. These genes were grouped into five clusters based on their expression

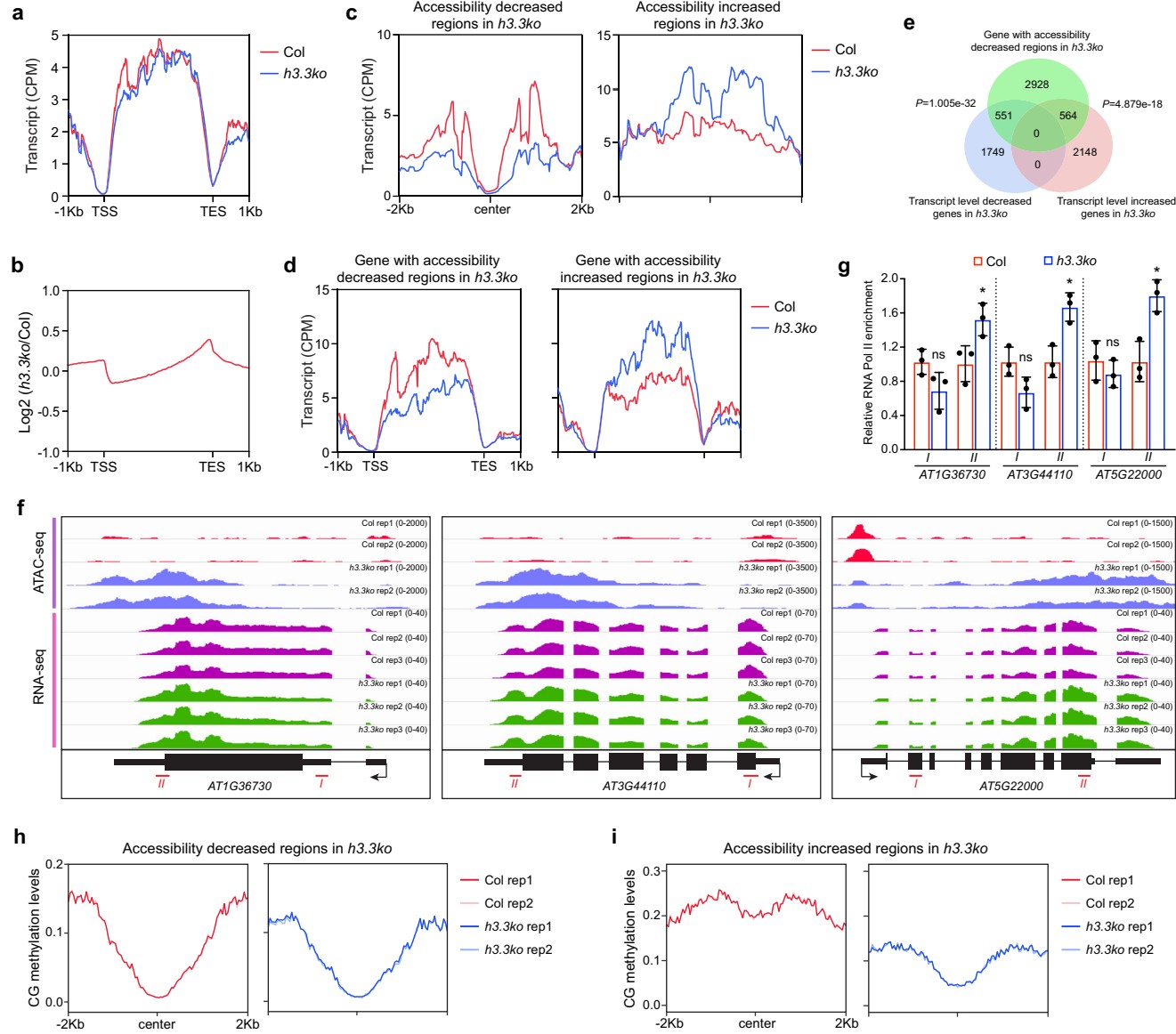

**Fig. 5 | Altered chromatin accessibility in *h3.3ko* correlates with changes in transcription and gene body DNA methylation. a** Metaplot of transcripts in Col and *h3.3ko* mature seeds over all genes. The profiles were generated after merging three biological replicates. **b** Metaplot of *h3.3ko* transcripts vs Col transcripts in mature seeds over all genes. The profiles were generated after merging three biological replicates. **c** Metaplot of transcripts in Col and *h3.3ko* mature seeds over accessibility significantly decreased and increased regions in mature *h3.3ko* seeds compared with Col. The profiles were generated after merging three biological replicates. **d** Metaplot of transcripts in Col and *h3.3ko* mature seeds over genes with accessibility significantly decreased and increased regions in mature *h3.3ko* seeds compared with Col. The profiles were generated after merging three biological replicates. **e** Venn diagrams of accessibility significantly decreased region-associated genes in mature *h3.3ko* seeds and transcript level significantly decreased

and increased genes in mature *h3.3ko* seeds. *P* values are based on the hypergeo-metric test. **f** Genome browser view of ATAC-seq and RNA-seq signals in Col and *h3.3ko* mature seeds over a few representative loci showing increased chromatin accessibility and transcript levels around the 3′ gene end in *h3.3ko* compared with Col. Red lines indicate regions examined by ChIP-qPCR. **g** Relative RNA Pol II enrichment levels determined by ChIP-qPCR at *AT1G36730*, *AT3G44110*, and *AT5G22000* in mature seeds. RNA Pol II enrichment levels were first normalized to input and then to that of Col. Values are means ± SD of three biological replicates. Statistical significance was determined by two-tailed Student's *t* test (*$P < 0.05$; ns, $P > 0.05$). **h, i** Metaplot of CG methylation levels in Col and *h3.3ko* mature seeds over accessibility significantly decreased (**h**) and increased (**i**) regions in mature *h3.3ko* seeds compared with Col.

patterns during imbibition in WT (Fig. 6d). Importantly, in *h3.3ko*, most of the genes in clusters 1–4 failed to fully activate during imbibition, while those in cluster 5 were not sufficiently repressed (Fig. 6d). GO analysis of these genes revealed mainly responsive processes and post-embryonic development (Fig. 6e). Hence, H3.3-dependent chromatin accessibility at these responsive and developmental genes is likely necessary for their transcriptional control during germination.

At regions where the accessibility was repressed by H3.3 in mature seeds (mainly localized at gene body and 3′ gene end), H3.3 remained deposited in WT during imbibition and in seedlings and is required to

maintain their low chromatin accessibility and to repress cryptic transcription at these regions (Fig. 6f and Supplementary Fig. 12c–e), indicating a canonical H3.3 distribution pattern around the 3′ gene end that constantly prevents chromatin opening and cryptic transcription at genic regions.

## Discussion

In *Arabidopsis*, both paternal and maternal-derived H3.3 is removed from the zygote genome a few hours after fertilization, followed by de novo synthesis of H3.3 before the zygote divides. This process was

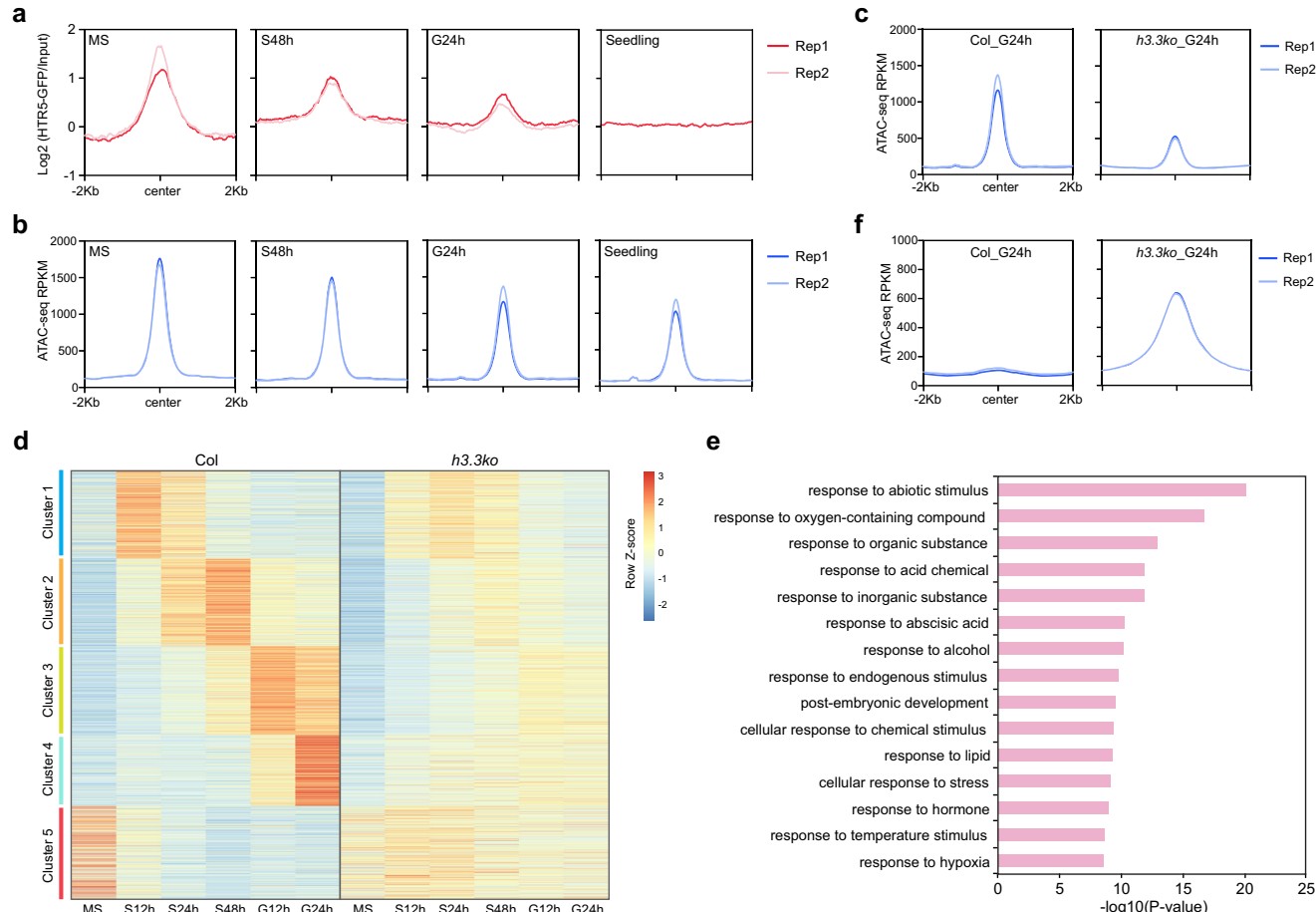

**Fig. 6 | Loss of chromatin accessibility in *h3.3ko* is associated with gene misregulation during germination. a, b** Metaplot of HTR5-GFP ChIP-seq (a) and ATAC-seq (b) signals in mature seeds, during imbibition and in the 10-day-old seedlings over accessibility decreased regions in mature *h3.3ko* seeds compared with Col. **c.** Metaplot of ATAC-seq signals in imbibed Col and *h3.3ko* seeds over accessibility significantly decreased regions in mature *h3.3ko* seeds compared with Col. **d** Heatmap showing expression profiles (z-normalized) of genes associated with accessibility significantly decreased regions in mature *h3.3ko* seeds compared with Col. Genes were grouped into five clusters based on their expression patterns during imbibition in Col. **e** GO analysis of genes associated with accessibility significantly decreased regions in mature *h3.3ko* seeds compared with Col. Top 15 representative terms are listed and ranked by *P* value, which was calculated with hypergeometric test. **f** Metaplot of ATAC-seq signals in imbibed Col and *h3.3ko* seeds over accessibility significantly increased regions in mature *h3.3ko* seeds compared with Col.

speculated to prevent the transmission of epigenetic memory from parents and meanwhile establish the zygotic epigenome critical for embryogenesis[42,59,60]. Interestingly, our genetic evidence suggests that H3.3 is not essential for early plant embryonic development. It is likely that H3.1 or other H3 proteins largely compensate for the loss of H3.3 at this stage, especially considering that H3.1 is abundantly produced during cell division, a dominant event in early embryogenesis[61]. After pattern formation at the torpedo stage, the embryo gradually ceases cell division and initiates maturation[2]. This is coupled with a strong reduction in H3.1 expression and an increase in H3.3 expression (Fig. 3d), which coincident with the extensive H3.1 replacement by H3.3 during embryo maturation[62]. Thus, H3.3 likely starts to be critical at embryo maturation stages, and reprograms the embryo epigenome for the establishment of post-embryonic developmental potentials. Consistently, embryos before torpedo stages are not capable to germinate when excised from seeds, while later stage embryos could germinate[2]. This role of H3.3 is comparable to the function of a specialized form of non-replicative H3 variant H3.10 in sperm that establishes chromatin accessibility at promoters of genes expressed in early embryogenesis[9,59]. It is unknown by which mechanism H3.3 is deposited to the 5′ gene ends in seeds but removed during germination and seedling development. This possibly involves the H3.3 deposition chaperone HIRA[63] that plays a moderate function in seed germination[64], or ATRX involved in the deposition of H3.3[65,66]. In

addition, specific chromatin remodelers might recruit H3.3 deposition machinery. Alternatively, removal of H3.1 from this region during embryo maturation may allow H3.3 to accumulate.

Several pieces of evidence suggest that seed germination is largely programmed during seed maturation[67]. Protein synthesis from stored mRNA in mature seeds is required for the completion of germination[68–70]. Moreover, mRNA abundance changes are observed within 1–2 h after imbibition without de novo protein synthesis required, highlighting the function of seed-stored transcription machinery in quickly adjusting transcription[71]. Our results suggest that H3.3 deposited in seeds is vital for their acquisition of germination capacity. Loss of H3.3 causes reduced and increased chromatin accessibility at 5′ and 3′ gene ends respectively, and results in impaired regulation of responsive and post-embryonic development genes and increased cryptic transcription. Since H3.3 is part of the chromatin core (nucleosome), it is unlikely that H3.3 impacts on other physiological processes that would in turn affect chromatin accessibility. Therefore, we propose the following model (Fig. 7): H3.3 is loaded at the 5′ gene ends/promoters in mature seeds, facilitating chromatin opening (Figs. 3f, g and 4). During germination and subsequent seedling development, 5′ enriched H3.3 is gradually lost. However, the chromatin accessibility is still maintained (Fig. 6). A plausible explanation is that the 5′ enriched H3.3 may initiate chromatin opening to enable transcription factor binding in seeds and during germination.

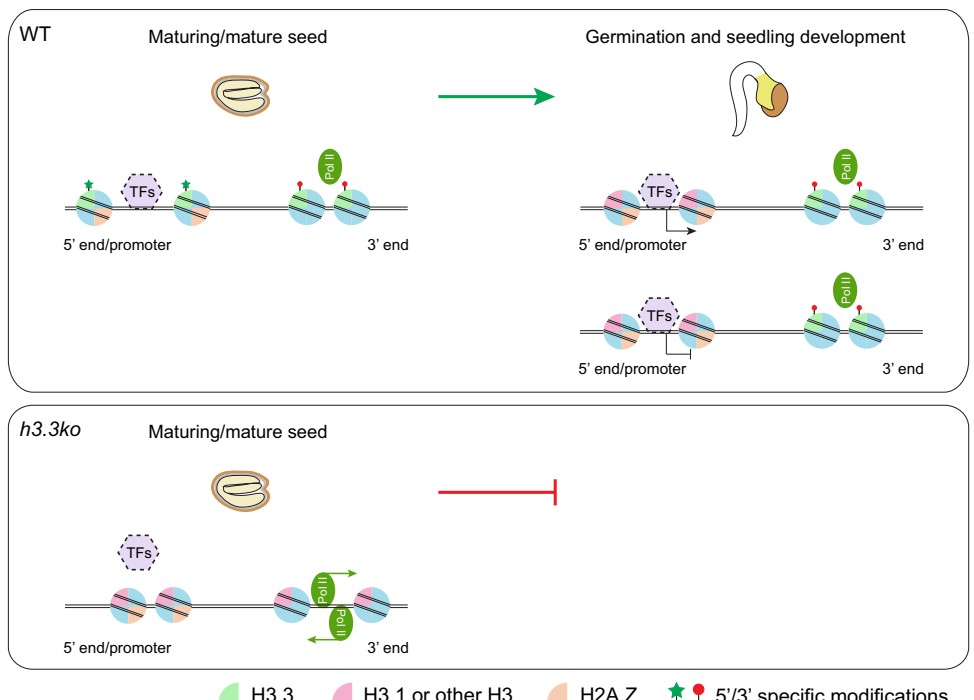

**Fig. 7 | A proposed function of H3.3 in chromatin accessibility regulation.** In maturing and mature seeds, H3.3 is loaded at the 5′ gene ends/promoters and establishes chromatin accessibility. During germination and seedling development, H3.3 is gradually removed from the 5′ gene ends, but the accessibility established by H3.3 is largely maintained probably by the binding of transcription factors (as indicated by the dashed border), and this may allow proper gene transcriptional regulation (activation or repression) during post-embryonic development. On the other hand, H3.3 is constantly deposited around the 3′ gene ends and prevents chromatin opening and cryptic transcription. Loss of H3.3 causes reduced and increased chromatin accessibility at 5′ and 3′ gene ends respectively, and results in compromised transcriptional regulation and severely impaired post-embryonic development. H2A.Z and the 5′ and 3′ specific histone modifications may also contribute to the H3.3-mediated chromatin accessibility control.

Afterward, the binding of transcription factors could sustain chromatin accessibility without H3.3. We found that the H3.3-established open chromatin regions are enriched with the transcription factor binding motifs especially those of bZIP, bHLH and BZR/BES family members (Supplementary Fig. 13 and Supplementary Data 4). On the other hand, H3.3 is constantly deposited around the 3′ gene ends and prevents chromatin accessibility and cryptic transcription. Nevertheless, it is still unclear whether the altered chromatin accessibility in *h3.3ko* is the direct cause of transcription changes and the impaired post-embryonic development, and the exact role of H3.3 in chromatin regulation and post-embryonic development remains to be investigated.

As a replacement histone, H3.3 deposition has been proposed to alter the nucleosome property[4,12]. In human cells, the presence of H3.3 destabilizes nucleosome, especially when co-existed with H2A.Z, leading to a nucleosome-free region at the 5′ gene end for transcription factor binding[49,72]. In mouse embryonic stem cell, H3.3 and H2A.Z collaborate around gene transcription start sites to set up a poised chromatin state for gene activation during cell differentiation[73]. Despite a predominant localization of H3.3 at the 3′ gene end in *Arabidopsis* vegetative tissues[32,33,35], our findings show that at certain developmental stages it is highly enriched around the 5′ gene end and facilitates chromatin opening. With fully developed leaves that have ceased cell division and expansion, a small fraction of H3.3 is also detected at the 5′ gene ends/promoters[74]. Besides in mature embryo, a considerable amount of H3.3 is deposited to replace H3.1 in plant cells that have exited proliferation and become terminal differentiated[62]. This implies a broad role of H3.3 in reprogramming the epigenome for cell fate determination. Further study of detailed mechanisms might provide a potential to manipulate chromatin accessibility and cellular differentiation by engineered H3.3 deposition.

## Methods

### Plant materials and growth conditions

*Arabidopsis h3.3ko* and HTR5-GFP lines were described previously[32,34]. Plants were grown in long-day conditions (16 h light/8 h dark) at 22 °C. For seed germination, after-ripened mature seeds (seeds harvested and stored at room temperature for three months) were sown on 1/2 MS medium. After being kept at 4 °C for two days (stratification), seeds were subjected for germination in long days at ~22 °C. Three replicates were performed to assess the germination and growth of *h3.3ko*, at least 110 seeds were used for each replicate. *h3.3ko* seeds germinated within two months after sowing were considered as germinated, and the rest seeds were discarded. The germination date of the first 12 germinated seeds was scored.

For dissected embryo growth analysis, embryos from after-ripened Col and *h3.3ko* seeds were isolated by manually removing endosperm and testa. The isolated embryos were cultivated on 1/2 MS medium under long-day conditions.

### Plasmid construction and *h3.3ko* complementation

For *h3.3ko* complementation constructs, *HTR5* or *HTR13* coding sequences were placed under the control of *HTR5* or *HTR13* promoter in the binary vector pBlligatorR43, which contains a mature seed stage-specific mCherry fluorescence selection marker[36]. For the estradiol induction, *HTR5* coding sequences were placed under the *pER8* promoter in the binary vector pMDC7[43,75]. To express *HTR5* with *at2S3* or *H2B.S* promoter, the *at2S3* or *H2B.S* promoter sequences were first cloned into pGWB510[76], followed by the insertion of *HTR5* coding sequences.

For the germination test of *h3.3ko;pER8::HTR5* seeds with or without estrogen (10 μM) treatment during imbibition, 50 seeds were assessed in each replicate and seeds were transferred onto a new 1/2

MS plate supplemented with or without estrogen every 5 days in case of estrogen degradation. The germination rates were scored after 2 month of imbibition. For the germination test of *h3.3ko;pat2S3::HTR5* and *h3.3ko;pH2B.S.::HTR5* seeds, germination rates were scored after 1 month and 10 days imbibition respectively. At least 38 seeds were assessed in each replicate.

## Seed size measurement
Complete dried mature seeds were spread on a white background. After the image taken, seed size was measured with ImageJ. At least 47 seeds were scored for each genotype.

## Seed storage protein analysis
One hundred and fifty Col or *h3.3ko* mature seeds were homogenized in protein extraction buffer (100 mM Tris-HCl pH 8.0, 0.5% SDS, 10% glycerol, 2% β-ME). Extracted proteins were resolved on 15% SDS-PAGE gel and stained with Coomassie Blue.

## RNA-seq
Total RNA was extracted from mature seeds (seeds stored for three months after harvesting) or imbibed seeds using Minibest plant RNA extraction kit (Takara). Three independent biological replicates were performed. Sequencing libraries were prepared with BGISEQ-500 RNA-seq library preparation kit according to the manufacturer's instruction. For strand-specific RNA-seq, total RNA extracted from Col and *h3.3ko* mature seeds were subjected to library preparation with MGIEasy RNA directional library preparation kit. Prepared libraries were sequenced on a DNBSEQ-T7 platform and paired-end 150 bp reads were generated.

## RNA-seq data analysis
Adapter trimming was performed and low-quality reads were filtered with fastp version 0.20.1[77]. Reads were mapped to the *Arabidopsis* genome (TAIR10) using Hisat2 version 2.1.10[78]. The numbers of mapped reads were determined with SAMtools version 1.9[79], and are listed in Supplementary Table 1. Reads per gene were counted by HTseq version 0.11.2[80]. Transcripts per million (TPM) values were generated using R. Differential gene expression analysis was performed using DESeq2 version 1.26.0[81]. Genes that displayed a more than two-fold expression change and had a *P* adjust value <0.05 were considered as differentially expressed. For data visualization, bigwig coverage files were generated using deepTools utility bamCoverage[82] with a bin size of 10 bp and normalized with CPM. The PCA plot was generated with plotPCA in DESeq2 version 1.26.0. The expression of *HTR4*, *HTR5* and *HTR8* during embryogenesis was extracted from the published datasets[45]. Gene ontology analysis was performed with DAVID (https://david.ncifcrf.gov)[83]. Clustering of genes associated with accessibility decreased regions in mature *h3.3ko* seeds compared with Col was performed based on their expression changes during germination in Col using K-Means clustering in R.

## RT-qPCR
Total RNA was extracted from mature seeds (seeds stored for three months after harvesting), imbibed seeds or 7-day-old seedlings using Minibest plant RNA extraction kit (Takara). Reverse transcription was performed using HiScript III 1st Strand cDNA Synthesis Kit (Vazyme, R312-02). Real-time quantitative PCR was conducted on an Applied Biosystems QuantStudio 6 Flex Real-Time PCR System using ChamQ Universal SYBR qPCR Master Mix (Vazyme, Q711-02). *PP2A* was used as an endogenous control for normalization. Three independent biological replicates were performed for each line and condition. Primers used to amplify *HTR5, HTR13, GA3OX1, GA3OX2, GA20OX1, GA20OX2, CYP707A2,* and *PP2A* are listed in Supplementary Table 2. The *HTR5* amplification primers were designed to only amplify the WT but not Crispr-mutated *HTR5*. Primers amplifying *HTR5* and *HTR13* were tested to have similar amplification efficiency.

## ChIP-seq
ChIP was performed with mature HTR5-GFP seeds (seeds stored for three months after harvesting) or seeds imbibed for different times. Materials were ground with liquid nitrogen into fine powder and immediately fixed with 1% formaldehyde[32,33,35,50–52]. Nuclei were extracted and mononucleosomes were generated with micrococcal nuclease (MNase) (Sigma, N5386) digestion as previously described[52]. Immunoprecipitation was conducted with anti-GFP (Thermo Fisher Scientific, A-11122) or anti-H2A.Z[51] antibody. After antibody incubation, Protein A Dynabeads (Thermo Fisher Scientific, 10002D) were added to collect the immunocomplexes. After washing, elution, reverse cross-linking and DNA purification, recovered DNA was subjected to library preparation with VAHTS universal DNA library prep kit for Illumina (Vazyme, ND607) according to the manufacturer's instruction and sequenced with Illumina NovaSeq 6000 to generate paired-end 150 bp reads. ChIP-seq experiments were performed with two independent biological replicates.

## ChIP-seq data analysis
Adapter trimming was performed and low-quality reads were filtered with fastp version 0.20.1[77]. Reads were mapped to the *Arabidopsis* genome (TAIR10) with Bowtie2 version 2.4.2[84], and filtered for duplicated reads by using Picard version 2.24.0 MarkDuplicates (https://github.com/broadinstitute/picard). The numbers of mapped reads were determined with SAMtools version 1.9[79], and are listed in Supplementary Table 1. For data visualization, bigwig coverage files were generated using deepTools utility bamCoverage[82] with a bin size of 10 bp and normalized to input or sequencing depth using RPKM. Visualization was performed with IGV version 2.7.2[85]. Heatmaps and average normalized ChIP-seq profiles were generated using deepTools utilities plotHeatmap and plotProfile. Published HTR5-GFP ChIP-seq data in seedlings were processed in the same manner[35].

## ChIP-qPCR
Mature seeds were ground with liquid nitrogen into fine powder and immediately fixed with 1% formaldehyde. Chromatin was sheared by sonication and immunoprecipitation was performed with anti-RNA Pol II antibody (abcam, ab26721). The amounts of immunoprecipitated DNA were quantified by real-time quantitative PCR. Three independent biological replicates were performed. Primers used for PCR amplification are listed in Supplementary Table 2.

## ATAC-seq
Mature seeds (seeds stored for three months after harvesting), seeds imbibed for different times or 10-day-old seedlings were chopped in lysis buffer (15 mM Tris-HCl pH 7.5, 20 mM NaCl, 80 mM KCl, 0.5 mM Spermine, 0.2% Triton X-100, 5 mM β-ME, protease inhibitor cocktail). After being filtered with a 30μm filter (CellTrics), nuclei were stained with DAPI and subjected to fluorescence-activated cell sorting (FACS). 50,000 nuclei per sample were collected and mixed with Tn5 transposase (Mei5bio, MF650-01), after 20 min of incubation at 37 °C, tagmented DNA was recovered with ChIP DNA Clean & Concentrator kit (ZYMO, D5205). After PCR amplification, sequencing libraries were purified with VAHTS DNA Clean Beads (Vazyme N411) and sequenced with Illumina NovaSeq 6000 to generate paired-end 150 bp reads. ATAC-seq experiments were performed with two independent biological replicates.

## ATAC-seq data analysis
Adapter trimming was performed and low-quality reads were filtered with fastp version 0.20.1[77]. Reads were mapped to the *Arabidopsis* genome (TAIR10) with Bowtie2 version 2.4.2[84]. Reads mapped to chloroplast and mitochondria genome were removed and duplicated reads were filtered with Picard version 2.24.0 MarkDuplicates (https://github.com/broadinstitute/picard). The numbers of mapped reads were

determined with SAMtools version 1.9[79], and are listed in Supplementary Table 1. ATAC-seq open chromatin peaks were identified using MACS2 version 2.1.2 with default parameters[86]. The q-value cutoff for peak calling was 0.05. Only peaks identified from both biological replicates were kept. Peak distributions were analyzed with ChIPseeker version 1.22.1[87]. Promoter refers to accessible regions located <2 kb upstream of the transcription start sites (TSS), downstream refers to accessible regions located <1 kb downstream of the transcription end sites (TES), Intergenic region refers to accessible regions located >2 kb upstream of TSS and more than 1 kb downstream of TES. To identify differentially enriched peaks, a common peak set was created by merging peaks in Col and *h3.3ko*, scores for open chromatin peak regions were calculated with deepTools utility multiBigwigSummary[82], and differential peaks were called by requiring more than two-fold difference and *P* adjust <0.05. Genes were assigned to the peaks when the genic and 1 kb upstream regions overlapped with the peak regions for at least one base pair. For data visualization, bigwig coverage files were generated using deepTools utility bamCoverage[82] with a bin size of 10 bp and normalized with RPKM. DNA motif analysis was performed with HOMER (version 4.11) using "findMotifs.pl"[88].

## BS-seq

Genomic DNA was extracted from mature seeds (seeds stored for three months after harvesting) with Quick-DNA Plant/Seed Miniprep Kit (ZYMO, D6020). 200 ng fragmented genomic DNA (200 bp–500 bp) was used for library preparation with VAHTS universal Pro DNA library prep kit for Illumina (Vazyme, ND608). NEBNext Multiplex Oligos for Illumina (NEB, E7535) were used for adapter ligation. After purification with VAHTS DNA Clean Beads (Vazyme N411), bisulfite conversion was performed with EZ DNA Methylation-Gold Kit (ZYMO, D5005), followed by DNA purification and PCR amplification. Libraries were sequenced with Illumina NovaSeq 6000 to generate paired-end 150 bp reads. BS-seq experiments were performed with two independent biological replicates.

## BS-seq data analysis

Adapter trimming was performed and low-quality reads were filtered with fastp version 0.20.1[77]. Reads were mapped to the *Arabidopsis* genome (TAIR10) with BS-Seeker2 version 2.1.8 using default parameters[89]. Duplicated reads were filtered with Picard version 2.24.0 MarkDuplicates. The numbers of mapped reads are listed in Supplementary Table 1. CG, CHG, and CHH methylation levels were calculated with CGmaptools (version 0.1.2)[90]. Methylation data were visualized with ggplot2 in R (version 1.1.1106). CG DMRs were called with CGmaptools using dmr function with default parameters.

## Statistical analysis

Statistical tests performed are indicated in the figure captions.

## Reporting summary

Further information on research design is available in the Nature Portfolio Reporting Summary linked to this article.

## Data availability

The datasets generated during the current study are available in the GEO repository GSE209645. The ChIP-seq data of HTR5-GFP in seedlings were downloaded from GSE167384. Source data are provided with this paper.

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

## Acknowledgements

We thank Dr. Hidenori Takeuchi for providing CRISPR-Cas9 generated *htr4;htr5* double mutant. Work in D.J. laboratory was supported by the Strategic Priority Research Program of the Chinese Academy of Sciences (Precision Seed Design and Breeding, XDA24020303), the National Natural Science Foundation of China (31970527, 32170545, and 32150610472) and the National Key R&D Program of China (2019YFA0903903). Work in F.B. laboratory was supported by FWF grant (P30802).

## Author contributions

F.B. and D.J. conceived the research. T.Z., J.L., M.X., J.P., L.M., and D.J. performed experiments. H.Z. performed bioinformatics analysis. F.B. and D.J. wrote the paper with inputs from all co-authors.

## Competing interests

The authors declare no competing interests.
