## [Peer Review File · Nature Communications]

Reviewers' Comments:

Reviewer #1:

Remarks to the Author:

In this manuscript, the authors studied the role of Arabidopsis histone variant H3.3 in seed germination and post-embryogenesis. The authors focused on the mature embryo-specific 5' gene end distribution of H3.3 and its relationship with chromatin accessibility and gene transcription. It is interesting that the pioneer role of H3.3 in initiating chromatin opening at regulatory regions in mature embryo might license the embryonic to post-embryonic transition. However, a major flaw of this manuscript is that the authors didn't perform detailed parallel and comparative analysis among the mature seeds, germinating seeds and seedlings regarding their transcriptomes, H3.3 profiles and chromatin accessibilities to reveal how H3.3 protein loaded to chromatin in mature seeds and dynamics of H3.3 during germination affect the processes of seed germination and post-embryogenesis. In addition, some of the data were omitted in the manuscript and need further experiments.

1. The *htr4;htr5;htr8/+* mutants have unchanged fruit set rates, but some seeds show a particularly delayed germination phenotype. What is the segregation ratio between normal seeds and those with delayed germination?
2. In Fig 1e, the expression levels should be also tested at the protein level by western blots.
3. Figure 2, the RNA-seq data should be validated by qRT-PCRs, Line 181, not "the expression level", should be transcription level. Moreover, it is also necessary to analyze and compare the changes of transcriptomes of mature seeds, germinating seeds and seedlings of H3.3ko mutant.
4. Lines 188-189, the word "responsive" appears twice, should be used more accurately.
5. Figure 4a, the authors claim that the overall whole-genome distribution patterns of accessible regions were similar in WT and h3.3ko, but in some regions, clear differences can be observed between WT and h3.3ko.
6. Figure 4b-d, in addition to the open chromatin of overall gene-coding regions shown here, the chromatin accessibility of several randomly selected individual genes should be shown.
7. Figure 5a, Overall, the distribution patterns of transcripts on genes were slightly altered in h3.3ko that the transcript levels around the 5' end were reduced, and that towards the 3' end were increased.
8. Figure 5a-b, "the distribution patterns of transcripts on genes were slightly altered in h3.3ko that the transcript levels around the 5' end were reduced, and that towards the 3' end were increased". Are there significant differences in these changes, the authors should give statistics analysis.
9. Lines 231-232, "3278 regions became less accessible in h3.3ko, while 3436 regions gained accessibility". How many of these up- and down-regulated regions are on the different regions of same genes and how many are on different genes?
10. During germination, gradual reduction of H3.3 at the 5' gene end, the overall chromatin accessibility at this region was largely maintained. The expression levels of genes with chromatin accessibility were not reduced and rather increased. The authors believe that "the mature embryo-specific 5' loading of H3.3 creates an open chromatin configuration, which is maintained during subsequent development to license gene expression in response to environmental stimuli and developmental signals". More experiments are needed to support the conclusion.
11. Explain that both expression down- and up-regulated genes in h3.3ko carry higher levels of H3.3, and the H3.3 distribution patterns at their loci were similar.

Reviewer #2:

Remarks to the Author:

This manuscript describes experiments aimed at comparing the role of Arabidopsis histone variant H3.3 in embryos and post-embryonic development. It is reported that H3.3 is not needed for embryo development but it is essential for seed germination. Authors find that H3.3 localizes in embryos preferentially to the 5' end of genes, in contrast with the situation in vegetative tissues, which correlates with chromatin accessibility. The major claim is that H3.3 has a pioneer role by initiating chromatin opening at regulatory regions in mature embryos.

Overall this is an interesting observation that adds on the knowledge of H3.3 role at various stages

of Arabidopsis development. The experimental approach is appropriate to deal with embryo lethal phenotype of the full knockout of the three H3.3 genes. Some of the conclusions are not sufficiently supported by experimental data (see below).

Comments

1. A general comment. It is challenging to understand how several thousands of genes are differentially expressed in the H3.3 mutant embryos, when they are apparently developed normally.
2. The text needs to be extensively improved. Many sentences are not correctly constructed and are difficult to follow. Even more important is that frequently the words and terms used are not appropriate or sufficiently precise to define what authors are talking about. One example: use of "downregulated peaks" is not appropriate (less accessible regions). Another: "expression down- and upregulated genes", but there are other cases of inappropriate terms.
3. Fig. 1f. This panel is too small to appreciate conveniently the details.
4. Fig 2. It is not clearly specified what is the source for the RNAseq experiments: whole mature seeds? purified embryos (as in Fig. 1f)? embryos obtained after seed imbibition?
5. Fig. 2c is not particularly informative. What can be deduced about the major gene regulatory networks affected, and necessary for germination? This would be a really key information, e.g. transcriptomic data at different times during germination comparing wt and mutant.
6. Fig. 2. It seems that ~25% of embryos initiate germination. What is their transcriptomic profile?
7. Fig 3b. The metaplots need to be accompanied by heatmaps where the contribution of outliers, if any, can be evaluated more properly.
8. Fig. 4. Again, clarify the source.
9. Fig. 4b, 4d. An apparent discrepancy occurs between the metaplots in 3b (H3.3) and 4b (ATAC) and the image in the browser (4d). Here, H3.3 is abundant upstream from the TSS, with a sharp decrease at the TSS. This is not observed in the metaplots, where the peak colocalizes almost perfectly with the TSS. The ATAC and H3.3 signals cover a good region (with several enriched peaks) upstream the TSS. This is not observed in the metaplots. Is this due to outliers? What is their contribution? A detailed explanation is needed.
10. Fig. 5. This set of data are quite confusing, or they are not described in sufficient detail. Fig. 5a, what are the reads upstream of the TSS and downstream of the TES coming from, with a tendency to increase? Are these reads derived from adjacent coding regions? If so, what is the meaning? This is not reflected in panel Fig. 5e.
11. Fig. 5e. Significance of the increasing amount of reads towards the 3'end of the gene. Does this correlate with increasing RNA Pol II occupancy? A validation by ChIP should shed light on this.
12. Fig. 6d (see also above, point 8). Again the location of peaks, upstream of TSS, does not match what is reported in the metaplots.
13. Fig. 6f. This in silico analysis needs to be evaluated in the light of the GO analysis, to identify gene regulatory networks primarily affected by the loss of H3.3. Otherwise, it is not very informative, beyond describing the presence of putative TFBS.

Reviewer #3:

Remarks to the Author:

This manuscript examines the impact of the absence of the histone variant H3.3 on both mature seed chromatin and subsequent plant development. As the authors note, replicative histone H3.1

and replacement histone H3.3 evolved separately in plants and animals. Each is associated with distinct chromatin states/capabilities in plants and animals, some shared and some not. Here the authors demonstrate that the absence of H3.3 in plants is associated with altered transcript representation and altered chromatin in mature seeds. Further, seeds lacking H3.3 exhibit substantially delayed germination and defective development. The spectrum of chromatin and developmental phenotypes of wild-type and H3.3-depleted seeds/seedlings is of substantial interest. As currently described, however, it is unclear what is cause and effect, and the presented model lacks key supporting data. Addressing the concerns that follow would meaningfully increase the impact of these studies.

The authors are strongly advised not to mix precedent from plants and animals without explicit attribution in every case. As the authors note, H3.1 and H3.3 evolved independently in each, and the authors appear to be describing a unique role for H3.3 in plants. Yet the authors frequently cite results obtained using animal systems to justify/motivate properties or proposed analyses in plants without making the reader aware that they are doing so unless the reader looks up the cited references. This is particularly the case in the introduction, but also occurs in the rest of the manuscript. Conflating results from distinct systems in this fashion impairs clarity and should be avoided to give the reader the opportunity to more easily assess both the rationale and the strength of the proffered arguments. A particular opportunity to address this point is at line 77 "higher plants acquired comparable features". Which features are these specifically, and which features are distinct in each?

Analysis of transcript levels of mature seeds is not "expression". These seeds are quiescent (pending clarification of how define "mature" see below). Thus what the authors are examining here is very distinct, for example, from characterizing chromatin and gene expression in rosette tissue. In this case, the authors are looking at transcript levels and chromatin status at some point after which plants have largely stopped modifying both. This situation is static rather than homeostatic. Further, no data are provided with regards to order of events here. The authors appear to assume that transcript levels and chromatin status reflect concurrent events when in reality it is unknown how the mature state was attained. Did actual transcription cease prior to deposition of H3.3? Similarly, does open chromatin reflect presence of H3.3 or the process of deposition of H3.3? In this regard, it is worth noting that the pattern of H3.3 enrichment in mature seeds is not predictive of increased or decreased transcript levels of genes. Similarly, the authors do not describe a correlation between increased or decreased transcript levels and chromatin accessibility as determined by ATAC-seq.

With regards to transcript analysis and linking observations together into some sort of model, it is surprising that the authors appear to switch from one definition of differential expression to another in Figure 5. Here the authors examine average reads over genes with altered chromatin accessibility at the 5' and 3' end. Although the data do suggest that loss of H3.3 leads to increased accumulation of mapped reads in the 3' end of expressed genes, the relationship between this observation and differentially accumulated transcripts in Figure 2 is not addressed.

Overall, the authors have an abundance of correlative observations but are lacking key experiments that strongly support their model. The authors note that H3.3 is enriched at the 5' and 3' ends of genes, which overlaps regions of chromatin accessibility in wild-type, and further observe loss of accessibility in the 5' ends of genes and increased accessibility in the 3' ends of genes in h3.3ko lines. They then propose "a direct impact of H3.3 on chromatin accessibility" (line 239) based on these data. To explain these opposite effects at different locations, they then examine H2A.Z enrichment, motivated by data from animals based on the provided references, and see that H2A.Z is enriched at the 5' ends of genes, as has previously been established in plants. A convincing follow-up experiment here would be analysis of chromatin accessibility of the mature seeds of plants lacking the H2A.Z-encoding genes HTA8, HTA9, and HTA11. In the absence of such data, the proposed "direct" role remains speculative, mechanistically uncertain, and refuted by opposing results at different ends of genes. Similarly, the authors have not compared the binding of transcription factors (it is not a surprise to observe that transcription factor binding sites are present in the promoters of genes) in wild-type versus H3.3ko lines or examined chromatin accessibility at the 5' ends of genes in H3.3ko seeds that do successfully germinate. Inclusion of any of these experiments would have tested a "necessity" relationship suggested by

their presently correlative data and significantly enhanced the basis for their proposed model.

Additional comments:

There are numerous grammatical errors which impair clear understanding of what is written. A number of these are addressed below.

The use of "pioneer" as a descriptive term seems inappropriate. "Pioneer" transcription factors are described as such because they have the ability to alter gene expression in the context of otherwise refractory chromatin. In this usage, chromatin is the template rather than the actor. Conferring "actor" status to H3.3 with the designation of "pioneer" impedes conceptual clarity. The authors are instead advised to use a distinct nomenclature that captures the foundational role of chromatin as the substrate upon which other factors (transcription factors and remodelers) act.

Line 39: The meaning of the following sentence is unclear as written: "At physiological level, embryos entering seed maturation stages have gradually obtained their germination capacity."

Line 63: please provide specific plant reference for histone H3 phylogeny.

Line 158: Use of "necessary" is preferred here rather than indispensable. Otherwise, this statement presumes the inability to identify suppressor mutations (for example), for which there is no evidence.

Line 161: Please define first use of "mature" for isolation of seeds here and again in Materials and Methods so that these experiments can be replicated.

Line 194: Please give number of histone genes, types, and some measure of extent of overrepresentation in place of "many histone genes".

Line 203: Suggest "underrepresented" or "depleted" in place of "deprived"

Line 209: Please specify using own RNA-seq data here and in Sup. Fig.3b

Line 268 and 271: Is Figure 5e supposed to be cited twice?

Line 273: The authors appear to begin addressing cryptic transcription here without directly addressing this point. Doing so here (in the context of posing a testable hypothesis that is not about antisense transcripts) would help to make this transition a bit easier for the reader.

Line 318 and following: It is unclear why the authors focus on the loss of the H3.3 peak at the 5' end and do not address the dramatic increase in H3.3 enrichment in gene bodies during germination in agreement with reference 14 from animals and the general concept of a replacement histone. The data certainly seem to warrant this observation.

Line 329: Suggest replacing "is completely depleted from" with "enrichment is not detected in"

Line 358: "before the"

Line 379: "this possibly involves"

Line 393: "that may facilitate"

Line 463: please define "mature" (e.g. age and treatment of seeds).

Line 468: Missing reference to Supp. Table 2 here to determine number of mapped reads.

Line 485: Please provide citation that demonstrates that freezing tissue and then treating with formaldehyde gives equivalent results to treating with formaldehyde and then freezing as is typically done. Reference 71 may state that same thing was done here, but would like to see citation with data that demonstrates that the order of these steps is exchangeable.

Line 498: Number of mapped reads in Supp. Table 2 are substantially below ENCODE guidelines. Please explicitly address in manuscript.

Line 512: How did authors "chop" mature seeds for isolation of nuclei? Were they hydrated before this? If yes, for how long? The chromatin is likely changing during this time.

Line 526: Number of mapped reads in Supp. Table 2 are substantially below ENCODE guidelines. Please explicitly address in manuscript.

Line 886: "Expressing" should be used in place of "expression"

References: 48 and 53 are repeats

Figure 1: Is there a citation providing data that the HTR13 promoter used is functional? Otherwise, the absence of an impact is easily interpreted as a consequence of using a non-functional promoter.

Figure 2b: Please specify if cited data are from embryo proper or entire seed.

Figure 2d and 2e: Please confirm that inset bar refers to log₂ fold change.

Figure 4c: Please specify how different regions are defined in Materials and Methods.

Figure 5e: Please specify what type of gene (up or down, 5' or 3') CTR1 is supposed to exemplify.

Figure 6: Key needs to be provided for designation of types of samples.

Figure 6d: Please specify what type of gene (up or down, 5' or 3') phyB is supposed to exemplify.

Figure 6e: Please confirm that these are wild-type genes. Does this include both 5' and 3'? Genes that go up and down? Unclear what readers should conclude from these data as presented.

Figure 7: As noted above, no data are presented that H2A.Z is necessary for open chromatin or that TFs do not bind in H3.3ko seedlings. In the absence of such data, this model seems speculative.

Reviewer #1 (Remarks to the Author):

In this manuscript, the authors studied the role of Arabidopsis histone variant H3.3 in seed germination and post-embryogenesis. The authors focused on the mature embryo-specific 5' gene end distribution of H3.3 and its relationship with chromatin accessibility and gene transcription. It is interesting that the pioneer role of H3.3 in initiating chromatin opening at regulatory regions in mature embryo might license the embryonic to post-embryonic transition. However, a major flaw of this manuscript is that the authors didn't perform detailed parallel and comparative analysis among the mature seeds, germinating seeds and seedlings regarding their transcriptomes, H3.3 profiles and chromatin accessibilities to reveal how H3.3 protein loaded to chromatin in mature seeds and dynamics of H3.3 during germination affect the processes of seed germination and post-embryogenesis. In addition, some of the data were omitted in the manuscript and need further experiments.

Response: We thank the reviewer for these constructive comments. We have now provided additional results to show that expressing H3.3 at the late seed maturation stage only could stimulate the germination of *h3.3ko*, while induction of H3.3 expression during imbibition only was not sufficient to induce the germination of *h3.3ko* (Figure 3a-3f). These results suggest that H3.3 in seeds is already critical for germination, though this is not to say that H3.3 expressed during germination and afterwards is not important. We therefore reasoned that the earliest time point that H3.3 becomes critical could be the late seed maturation stage since H3.3 is not required for seed formation. Given that the mature seed stage is the earliest time point when we could distinguish *h3.3ko* from WT, we mainly put our focus on this stage. We have also performed RNA-seq and ATAC-seq with imbibed *h3.3ko* seeds. And the results show that the chromatin accessibility at the 5' gene end in the imbibed *h3.3ko* remained much lower than that in WT (Figure 6c and Supplemental Figure S12b), considering that the H3.3 accumulation levels at these regions were gradually reduced during imbibition (Figure 3g and 3h and Figure 6a), this suggests that H3.3 in mature/maturing seeds are critical for the establishment of chromatin accessibility. In addition, transcriptome analysis in WT and *h3.3ko* indicates that this H3.3-established chromatin accessibility is important for gene transcriptional regulation during germination (Figure 6d).

1. The *htr4;htr5;htr8/+* mutants have unchanged fruit set rates, but some seeds show a particularly delayed germination phenotype. What is the segregation ratio between normal seeds and those with delayed germination?

Response: We thank the reviewer for raising this question. The percentage of seeds with delayed germination (*h3.3ko*) is around 7.5% to 11.5%. This ratio is lower than the expected 25% because knockout of H3.3 partially impairs male gametogenesis, and when crossing *htr4;htr5;htr8/+* (*h3.3ko/+*) with the wild-type Col mother, the transmission of *h3.3ko* was reduced to around 16% instead of the expected 50% (Wollmann et al., 2017, doi: 10.1186/s13059-017-1221-3). We have added this information in the revised manuscript (Line 114-121, Supplemental Figure S1b).

2. In Fig 1e, the expression levels should be also tested at the protein level by western blots.

Response: We thank the reviewer for this suggestion. These transgenic HTR5 (H3.3) and HTR13 (H3.1) are not fused with any tags, and unfortunately the plant H3.1 and H3.3 specific antibodies are currently not available. Therefore we were not able to assess their protein expression levels by western blot. Nevertheless, we examined the transcript levels of HTR5 and HTR13 in the mature seeds of transgenic lines. The HTR5 amplification primers were designed to only amplify WT/transgenic HTR5 but not the Crispr-mutated HTR5 in *h3.3ko*, and the HTR5 and HTR13 primers were tested to have similar amplification efficiency. The results showed that HTR13 driven by the HTR5 promoter was highly expressed in mature seeds, while HTR5 driven by the HTR13 promoter was expressed at low levels (Supplemental Figure S1d). The same HTR13 promoter was successfully used for the *h3.1kd* complementation (Jiang and Berger, doi: 10.1126/science.aan4965), and thus the low expression of HTR5 under the HTR13 promoter is likely because the HTR13 promoter is not active at the mature seed stage (also see Figure 3d).

3. Figure 2, the RNA-seq data should be validated by qRT-PCRs, Line 181, not “the expression level”, should be transcription level. Moreover, it is also necessary to analyze and compare the changes of transcriptomes of mature seeds, germinating seeds and seedlings of *H3.3ko* mutant.

Response: We thank the reviewer for these suggestions. We have modified “the expression levels” to “the transcript levels”. We have also profiled the transcriptome of WT and *h3.3ko* in mature seeds and imbibed/germinating seeds by RNA-seq (Figure 2). The results showed that compared with WT, the overall transcriptome changes in *h3.3ko* were less dynamic during imbibition. Some GA biosynthesis and ABA catabolic genes key for germination failed to activate in *h3.3ko* during imbibition (also validated by RT-qPCR in Supplemental Figure S2a), consistent with its impaired germination. In fact, the transcriptome difference between WT and *h3.3ko* was already obvious at the mature seed stage, and we have provided several data to show that defects in the *h3.3ko* mature seeds already affect germination (Figure 3a-3f). Therefore, the earliest stage that H3.3 becomes critical could be the mature seed stage (or the earlier seed maturation stage).

We did not profile the transcriptome of *h3.3ko* seedlings because: 1) only a few (~25%) *h3.3ko* seeds could germinate (within two months) and the majority of the germinated *h3.3ko* ceased development just after germination even without expanding cotyledons (figure 1c and 1f), making it hard to collect enough seedlings for RNA-seq. 2) Because the germination of *h3.3ko* is much delayed and not uniform compared with WT (Figure 1e), and after germination, the development of *h3.3ko* is much slower or stopped, it's not possible to define a WT seedling control that is considered to be at the same developmental stage as *h3.3ko* for comparison. 3) Given that the function of H3.3 in the mature seed stage seems already critical, we mainly focused to

investigate the function of H3.3 at this stage.

4. Lines 188-189, the word “responsive” appears twice, should be used more accurately.

Response: We thank the reviewer for pointing out this mistake. We have removed this description in the text based on other reviewers’ comments.

5. Figure 4a, the authors claims that the overall whole-genome distribution patterns of accessible regions were similar in WT and h3.3ko, but in some regions, clear differences can be observed between WT and h3.3ko.

Response: We thank the reviewer for raising this point. We have modified the description and only state that overall accessible regions were mainly enriched at euchromatin in both WT and *h3.3ko*.

6. Figure 4b-d, in addition to the open chromatin of overall gene-coding regions shown here, the chromatin accessibility of several randomly selected individual genes should be shown.

Response: We thank the reviewer for this suggestion. We have now added heatmaps to show the accessibility over all genes (Figure 4c and Supplemental Figure S5b and S5c), we have also included more examples to show the chromatin accessibility changes in *h3.3ko* (Figure 4e). The bigwig files have been deposited into the GEO database, and readers can download them and easily examine any loci with genome browser.

7. Figure 5a, Overall, the distribution patterns of transcripts on genes were slightly altered in h3.3ko that the transcript levels around the 5’ end were reduced, and that towards the 3’ end were increased.

Response: We assume this comment and the following one are the same question.

8. Figure 5a-b, “the distribution patterns of transcripts on genes were slightly altered in h3.3ko that the transcript levels around the 5’ end were reduced, and that towards the 3’ end were increased”. Are there significant differences in these changes, the authors should give statistics analysis.

Response: We appreciate the reviewer for raising this question. To perform statistical analysis, we divided the transcripts on genes into 50 bins from 5’ to 3’, and calculated the significance of the difference between Col and *h3.3ko* in each bin. As shown in Supplemental Figure S9b, the differences in the first two bins (5’) and last three (3’) bins are statistically significant.

9. Lines 231-232, “3278 regions became less accessible in h3.3ko, while 3436 regions gained accessibility”. How many of these up- and down-regulated regions are on the different regions of same genes and how many

are on different genes?

Response: We thank the reviewer for raising this question. We identified 4043 and 4388 genes that associate with accessibility decreased and increased regions in *h3.3ko* respectively, with 920 genes containing both (Line 339-341, Supplemental Figure S9c). Therefore, in most cases, genes are associated with only accessibility significantly decreased or increased region, while a small portion of them contains both.

10. During germination, gradual reduction of H3.3 at the 5' gene end, the overall chromatin accessibility at this region was largely maintained. The expression levels of genes with chromatin accessibility were not reduced and rather increased. The authors believe that "the mature embryo-specific 5' loading of H3.3 creates an open chromatin configuration, which is maintained during subsequent development to license gene expression in response to environmental stimuli and developmental signals". More experiments are needed to support the conclusion.

Response: We thank the reviewer for this comment. We have performed some additional experiments to show the importance of H3.3 in mature seeds (or maturing seeds): 1) we show that induction of H3.3 at the imbibition stage only is not sufficient to induce germination in *h3.3ko* (Figure 3a-3c), indicating that loss of H3.3 already caused strong defects in mature seeds that affect germination. 2) We found that expressing H3.3 with promoters only active at the late seed maturation stage could rescue the *h3.3ko* germination defects (Figure 3e and 3f), further supporting that seed expressed H3.3, which has a seed-specific 5' gene end enrichment pattern, is critical for germination. 3) We have performed ATAC-seq in the imbibed *h3.3ko* seeds, and the chromatin accessibility at the 5' gene end in the imbibed *h3.3ko* remained much lower than that in WT (Figure 6c and Supplemental Figure S12b). Considering that the H3.3 accumulation levels at these regions were gradually reduced during imbibition (Figure 3g and 3h and Figure 6a), this suggests that H3.3 in mature/maturing seeds are critical for the establishment of chromatin accessibility. 4) We have analyzed the expression of genes with accessibility decreased regions in *h3.3ko* during imbibition in both WT and *h3.3ko*. Most of the genes (cluster 1-4) were activated during imbibition in WT, but their activation was compromised in *h3.3ko*. In addition, genes in cluster 5 were not sufficiently repressed in *h3.3ko* during imbibition (Figure 6d). These results support that the H3.3-established chromatin accessibility is required for gene transcriptional regulation during germination.

We understand the reviewer's concerns regarding this claim. We have also tuned down our statement (Line 418-420).

11. Explain that both expression down- and upregulated genes in *h3.3ko* carry higher levels of H3.3, and the H3.3 distribution patterns at their loci were similar.

Response: We thank the reviewer for raising this question. We think it's because these expression down- and upregulated genes are mostly expressed genes (not expressed genes will not be identified as down- or upregulated), and H3.3 is more accumulated at expressed genes anyway. Therefore, these observations may not directly connect with the focus of this study, and we have removed these results in the revised manuscript.

Reviewer #2 (Remarks to the Author):

This manuscript describes experiments aimed at comparing the role of Arabidopsis histone variant H3.3 in embryos and post-embryonic development. It is reported that H3.3 is not needed for embryo development but it is essential for seed germination. authors find that H3.3 localizes in embryos preferentially to the 5' end of genes, in contrast with the situation in vegetative tissues, which correlates with chromatin accessibility. The major claim is that H3.3 has a pioneer role by initiating chromatin opening at regulatory regions in mature embryos.

Overall this is an interesting observation that adds on the knowledge of H3.3 role at various stages of Arabidopsis development. The experimental approach is appropriate to deal with embryo lethal phenotype of the full knockout of the three H3.3 genes. Some of the conclusions are not sufficiently supported by experimental data (see below).

Response: We thank the reviewer for these positive comments.

Comments

1. A general comment. It is challenging to understand how several thousands of genes are differentially expressed in the H3.3 mutant embryos, when they are apparently developed normally.

Response: We thank the reviewer for raising this point. Indeed, it is intriguing to find that the *h3.3ko* mature seeds showed no obvious morphological defects but strong transcriptome changes. We think likely H3.3 only starts to be essential from the late embryo maturation stage when the embryogenesis is largely done to prepare seeds for germination. We have added the results showing that expressing H3.3 with promoters only active at the late seed maturation stage could rescue the *h3.3ko* germination defects (Figure 3e and 3f and Supplemental Figure S4a), further supporting that seed expressed H3.3, which has a seed-specific 5' gene end enrichment pattern, is critical for germination.

2. The text needs to be extensively improved. Many sentences are not correctly constructed and are difficult to follow. Even more important is that frequently the words and terms used are not appropriate or sufficiently precise to define what authors are talking about. One example: use of “downregulated peaks” is not appropriate (less accessible regions). Another: “expression down- and upregulated genes”, but there are other cases of inappropriate terms.

Response: We appreciate the reviewer for this suggestion. We have tried our best to improve our writing.

3. Fig. 1f. This panel is too small to appreciate conveniently the details.

Response: We thank the reviewer for pointing out this issue. We have increased the size of Figure 1f in the revised manuscript.

4. Fig. 2. It is not clearly specified what is the source for the RNAseq experiments: whole mature seeds? purified embryos (as in Fig. 1f)? embryos obtained after seed imbibition?

Response: We thank the reviewer for raising this issue. We have now included both mature and imbibed seeds for the RNA-seq analysis, and we state in the manuscript that seeds stored for three months after harvesting (hereinafter referred to as mature seeds) were subjected to imbibition and RNA was extracted from seeds collected at various time points (Line 170-173, Figure 2a). Whole seeds (mature or imbibed) were also used for ChIP-seq, ATAC-seq and BS-seq. We did not isolate embryos for these experiments because: 1) mature Arabidopsis seed mainly contains embryo (endosperm in mature seed is only one layer of cells, and testa cells are dead), so the data we obtained should mainly reflect the situation in embryo. 2) Seed need to be imbibed for at least a short period for embryo isolation. In this case, the embryo from mature seed could not be isolated without imbibition and the isolated embryo after imbibition might not be exactly at the mature seed stage anymore. 3) Our embryo dissection experiment suggests that the defects of *h3.3ko* indeed are mainly in the embryo (Figure 1f).

5. Fig. 2c is not particularly informative. What can be deduced about the major gene regulatory networks affected, and necessary for germination? This would be a really key information, e.g. transcriptomic data at different times during germination comparing wt and mutant.

Response: We thank the reviewer for raising this point. We have now included transcriptome analysis at different imbibition time points (Figure 2). The results showed that misexpressed genes in *h3.3ko* during imbibition are enriched with responsive genes (Figure 2c). In addition, we found that some GA biosynthesis and ABA catabolic genes key for germination failed to activate in *h3.3ko* during imbibition (Figure 2d), consistent with its impaired germination. We also found that genes that lose their chromatin accessibility in mature *h3.3ko* seeds are enriched with responsive and post-embryonic development genes (Figure 6e), and these genes are not activated or repressed properly during imbibition in *h3.3ko* (Figure 6d).

6. Fig. 2. It seems that ~25% of embryos initiate germination. What is their transcriptomic profile?

Response: We thank the reviewer for raising this question. We apologize that although ~25% of *h3.3ko* seeds could germinate after prolonged imbibition (two months), we did not examine their transcriptome profile because: 1) If to collect seed materials before their germination, these 25% of *h3.3ko* seeds are not distinguishable from other *h3.3ko* seeds until they have germinated, and therefore we are not able to collect these 25% of seeds before germination to examine their transcriptome. 2) If to collect seedling materials

after their germination, the majority of the germinated *h3.3ko* immediately cease development even without expanding cotyledons (figure 1c and 1f), and their germination time is highly variable (Figure 1e). Thus it's hard to define a WT seedling control that is considered to be at the same developmental stage as these germinated *h3.3ko*. 3) In fact, the germination of these 25% of *h3.3ko* seeds is much delayed (the earliest one took about ten days to germinate) and their germinate time is highly variable (Figure 1e), we would expect that if a longer imbibition time (more than two months) was given, more *h3.3ko* seeds may germinate. Therefore, there seem to be phenotypic variations among these *h3.3ko* seeds. These 25% of *h3.3ko* seeds could be merely that happen to germinate within the two-month period we gave, but otherwise, they may not be significantly different from other *h3.3ko* seeds.

7. Fig 3b. The metaplots need to be accompanied by heatmaps where the contribution of outliers, if any, can be evaluated more properly.

Response: We thank the reviewer for this suggestion. We have now included heatmaps in Figure 3h and Supplemental Figure S4e.

8. Fig. 4. Again, clarify the source.

Response: In Figure 4, we used mature seeds for ATAC-seq. We have added this information in the text (Line 267-270).

9. Fig. 4b, 4d. An apparent discrepancy occurs between the metaplots in 3b (H3.3) and 4b (ATAC) and the image in the browser (4d). Here, H3.3 is abundant upstream from the TSS, with a sharp decrease at the TSS. This is not observed in the metaplots, where the peak colocalizes almost perfectly with the TSS. The ATAC and H3.3 signals cover a good region (with several enriched peaks) upstream the TSS. This is not observed in the metaplots. Is this due to outliers? What is their contribution? A detailed explanation is needed.

Response: We thank the reviewer for raising this point. We have now provided heatmaps for both H3.3 ChIP-seq and ATAC-seq. Indeed, heatmaps show that there are cases that H3.3 is localized at the promoter regions, and so do ATAC-seq signals (Figure 3h and 4c and Supplemental Figure S4e and S5b), and the metaplots show only average signals. This is also the case for ATAC-seq signals in seedlings (Lu et al., 2017, doi: 10.1093/nar/gkw1179). We have also added an example that the H3.3 and ATAC-seq signals were mainly localized around the 5' gene end (Figure 4e: AT1G72330).

10. Fig. 5. This set of data are quite confusing, or they are not described in sufficient detail. Fig. 5a, what are the reads upstream of the TSS and downstream of the TES coming from, with a tendency to increase? Are these reads derived from adjacent coding regions? If so, what is the meaning? This is not reflected in panel Fig. 5e.

Response: We thank the reviewer for raising this point. The *Arabidopsis* genome is quite compacted (~130Mb) and there are ~25000 genes. Therefore, genes can be close to each other, and the reads upstream of the TSS and downstream of TES are from adjacent genes. Please see below a randomly selected region of our RNA-seq data showing that no reads were detected at intergenic regions. The bigwig files are also deposited into the GEO database so that readers can download them and easily examine any loci with the genome browser.

11. Fig. 5e. Significance of the increasing amount of reads towards the 3' end of the gene. Does this correlate with increasing RNA Pol II occupancy? A validation by ChIP should shed light on this.

Response: We appreciate the reviewer for this suggestion. We selected a few genes showing increased chromatin accessibility and transcript levels around the 3' gene end and measured RNA Pol II enrichment levels at their loci by ChIP-qPCR. The results showed that the RNA Pol II enrichment levels were increased around the 3' gene ends (Figure 5f and 5g).

12. Fig. 6d (see also above, point 8). Again the location of peaks, upstream of TSS, does not match what is reported in the metaplots.

Response: We have added heatmaps together with metaplots (Figure 3h and 4c and Supplemental Figure S4e and S5b).

13. Fig. 6f. This in silico analysis needs to be evaluated in the light of the GO analysis, to identify gene regulatory networks primarily affected by the loss of H3.3. Otherwise, it is not very informative, beyond describing the presence of putative TFBS.

Response: We thank the reviewer for this suggestion. We have analyzed the GO terms of genes that associate with the H3.3-dependent open chromatin regions, and mainly found responsive processes and post-embryonic development (Figure 6e). We also compared their transcript levels in WT and *h3.3ko* during imbibition. Compared with WT, most of these genes could not be fully activated or repressed in *h3.3ko* during imbibition (Figure 6d), suggesting the importance of these H3.3-dependent open chromatin regions in the transcriptional regulation during germination.

Reviewer #3 (Remarks to the Author):

This manuscript examines the impact of the absence of the histone variant H3.3 on both mature seed chromatin and subsequent plant development. As the authors note, replicative histone H3.1 and replacement histone H3.3 evolved separately in plants and animals. Each is associated with distinct chromatin states/capabilities in plants and animals, some shared and some not. Here the authors demonstrate that the absence of H3.3 in plants is associated with altered transcript representation and altered chromatin in mature seeds. Further, seeds lacking H3.3 exhibit substantially delayed germination and defective development. The spectrum of chromatin and developmental phenotypes of wild-type and H3.3-depleted seeds/seedlings is of substantial interest. As currently described, however, it is unclear what is cause and effect, and the presented model lacks key supporting data. Addressing the concerns that follow would meaningfully increase the impact of these studies.

Response: We thank the reviewer for many constructive comments.

The authors are strongly advised not to mix precedent from plants and animals without explicit attribution in every case. As the authors note, H3.1 and H3.3 evolved independently in each, and the authors appear to be describing a unique role for H3.3 in plants. Yet the authors frequently cite results obtained using animal systems to justify/motivate properties or proposed analyses in plants without making the reader aware that they are doing so unless the reader looks up the cited references. This is particularly the case in the introduction, but also occurs in the rest of the manuscript. Conflating results from distinct systems in this fashion impairs clarity and should be avoided to give the reader the opportunity to more easily assess both the rationale and the strength of the proffered arguments. A particular opportunity to address this point is at line 77 “higher plants acquired comparable features”. Which features are these specifically, and which features are distinct in each?

Response: We thank the reviewer for this suggestion. We have modified the text and explicitly indicate in which system the results were obtained.

Analysis of transcript levels of mature seeds is not “expression”. These seeds are quiescent (pending clarification of how define “mature” see below). Thus what the authors are examining here is very distinct, for example, from characterizing chromatin and gene expression in rosette tissue. In this case, the authors are looking at transcript levels and chromatin status at some point after which plants have largely stopped modifying both. This situation is static rather than homeostatic. Further, no data are provided with regards to order of events here. The authors appear to assume that transcript levels and chromatin status reflect concurrent events when in reality it is unknown how the mature state was attained. Did actual transcription cease prior to deposition of H3.3? Similarly, does open chromatin reflect presence of H3.3

or the process of deposition of H3.3? In this regard, it is worth noting that the pattern of H3.3 enrichment in mature seeds is not predictive of increased or decreased transcript levels of genes. Similarly, the authors do not describe a correlation between increased or decreased transcript levels and chromatin accessibility as determined by ATAC-seq.

Response: We thank the reviewer for these comments. We agree that mature seeds are largely static. 1) Since we could not distinguish *h3.3ko* from WT until the mature seed stage because the seed development of *h3.3ko* and WT is morphologically identical, the earliest time point we could identify *h3.3ko* seeds is the mature seed stage (Line 240-245). Considering the H3.3 is not required for seed formation, it very likely starts to modulate chromatin accessibility from late seed maturation stage to regulate the production of RNA, which could be accumulated in mature seeds. In addition, the transcription in mature seeds is still active, albeit at a lower rate (Comai and Harada, DOI: 10.1073/pnas.87.7.2671) (Line 324-331). Therefore, we took the earliest possible stage that we could obtain (mature seeds) and profiled H3.3 accumulation, chromatin accessibility and transcriptome. 2) We have performed additional experiments to show that expressing H3.3 with promoters only active at the late seed maturation stage could rescue the *h3.3ko* germination defects, while inducing H3.3 expression during imbibition is not sufficient to promote germination in *h3.3ko* (Figure 3a-3f), further supporting the importance of H3.3 in seeds. 3) We have performed ATAC-seq with imbibed *h3.3ko* seeds (seeds not quiescent anymore). The results showed that the chromatin accessibility at the 5' gene end in the imbibed *h3.3ko* remained much lower than that in WT (Figure 6c and Supplemental Figure S12b). Considering that the H3.3 accumulation levels at these regions were gradually reduced during imbibition (Figure 6a), this suggests that H3.3 in mature/maturing seeds is critical for the establishment of chromatin accessibility. 4) We have added RNA-seq analysis with imbibed *h3.3ko* seeds and examined the expression of genes with accessibility decreased regions in *h3.3ko* during imbibition in both WT and *h3.3ko*. Most of the genes (clusters 1-4) were activated during imbibition in WT, but their activation was compromised in *h3.3ko*. In addition, genes in cluster 5 were not sufficiently repressed in *h3.3ko* during imbibition (Figure 6d). These results support that the H3.3-established chromatin accessibility is required for gene transcriptional regulation during germination. For accessibility increased regions in mature *h3.3ko* seeds, H3.3 remained accumulated during imbibition and the accessibility and transcript level at these regions remained higher in the imbibed *h3.3ko* seeds compared with WT (Figure 6f and Supplemental Figure S12b to S12e), suggesting that H3.3 constantly represses chromatin accessibility at these regions.

With regards to transcript analysis and linking observations together into some sort of model, it is surprising that the authors appear to switch from one definition of differential expression to another in Figure 5. Here the authors examine average reads over genes with altered chromatin accessibility at the 5' and 3' end. Although the data do suggest that loss of H3.3 leads to increased accumulation of mapped reads in the 3' end of expressed genes,

the relationship between this observation and differentially accumulated transcripts in Figure 2 is not addressed.

Response: We thank the reviewer for raising this point. In Figure 5, we show that genes with increased chromatin accessibility in *h3.3ko* overall had more transcripts towards the 3' gene end (Figure 5d). However, reads at these genes were unevenly distributed in *h3.3ko* and some of them were produced from antisense (Supplemental Figure S10), indicating that the levels of real meaningful full-length transcripts of these genes may not be much increased in *h3.3ko* (please also see examples in Figure 5f). On the other hand, the differential gene expression analysis uses total reads counted from the whole gene loci to identify transcript level increased and decreased genes (Figure 2e), transcript level increased genes (directly or indirectly affected by the loss of H3.3) identified in this way may have reads increased over whole gene loci or only around the 3' gene end. Therefore, we did not directly compare these two sets of genes, and we showed reads over genes in Figure 5 to avoid misleading.

Overall, the authors have an abundance of correlative observations but are lacking key experiments that strongly support their model. The authors note that H3.3 is enriched at the 5' and 3' ends of genes, which overlaps regions of chromatin accessibility in wild-type, and further observe loss of accessibility in the 5' ends of genes and increased accessibility in the 3' ends of genes in *h3.3ko* lines. They then propose “a direct impact of H3.3 on chromatin accessibility” (line 239) based on these data. To explain these opposite effects at different locations, they then examine H2A.Z enrichment, motivated by data from animals based on the provided references, and see that H2A.Z is enriched at the 5' ends of genes, as has previously been established in plants. A convincing follow-up experiment here would be analysis of chromatin accessibility of the mature seeds of plants lacking the H2A.Z-encoding genes HTA8, HTA9, and HTA11. In the absence of such data, the proposed “direct” role remains speculative, mechanistically uncertain, and refuted by opposing results at different ends of genes. Similarly, the authors have not compared the binding of transcription factors (it is not a surprise to observe that transcription factor binding sites are present in the promoters of genes) in wild-type versus *H3.3ko* lines or examined chromatin accessibility at the 5' ends of genes in *H3.3ko* seeds that do successfully germinate. Inclusion of any of these experiments would have tested a “necessity” relationship suggested by their presently correlative data and significantly enhanced the basis for their proposed model.

Response: We thank the reviewer for these suggestions. We have now included the ATAC-seq analysis with a hypomorphic *h2a.z* mutant (this is so far the strongest mutant we could obtain). The results showed that overall chromatin accessibility was slightly reduced at the 5' gene end, while chromatin accessibility at the 3' gene end was not changed compared with WT (Supplemental Figure S8a). Importantly, regions showed reduced accessibility in *h3.3ko* also lost accessibility in *h2a.z* (Supplemental Figure

S8b), but the loss of H2A.Z did not induce the chromatin accessibility at regions with increased accessibility in *h3.3ko* (Supplemental Figure S8c). In addition, we observed moderately delayed germination of *h2a.z* compared with WT (Supplemental Figure S8d). The accessibility changes in the *h2a.z* mutant were less strong compared with that in *h3.3ko*. This could be because *HTA9* is still expressed at low levels in this *h2a.z* mutant (Coleman-Derr and Zilberman, doi:10.1371/journal.pgen.1002988), or site-specific histone modifications enriched at the 5' or 3' gene end may also contribute to the accessibility control together with H3.3. We have discussed and included these possibilities in the text and Figure 7. To be precise, we have also removed the claim that H3.3 may directly regulate chromatin accessibility.

For the transcription factor (TF) binding analysis, we searched for the native antibodies of the *Arabidopsis* TFs so that we could directly compare TF binding in WT and *h3.3ko*. Plant TF antibodies are much less available than that of animals, and we eventually got antibodies for three TFs that could be verified by our western blot in seeds. Unfortunately, none of them could work in our ChIP experiments. We apologize that we could not compare TF binding in WT and *h3.3ko* at this point, and we have moved the identification of TF binding sites to the discussion section.

We apologize that although ~25% of *h3.3ko* seeds could germinate after prolonged imbibition (two months), we could not examine their chromatin accessibility because: 1) If to collect seed materials before their germination, these 25% of *h3.3ko* seeds are not distinguishable from other *h3.3ko* seeds until they have germinated, and therefore we are not able to collect these 25% of seeds before germination to examine chromatin accessibility. 2) If to collect seedling materials after their germination, the majority of the germinated *h3.3ko* immediately cease development even without expanding cotyledons (figure 1c and 1f), and their germination time is highly variable (Figure 1e). Thus it's hard to define a WT seedling control that is considered to be at the same developmental stage of these germinated *h3.3ko*. 3) In fact, the germination of these 25% of *h3.3ko* seeds is much delayed (the earliest one took about ten days to germinate) and their germinate time is highly variable (Figure 1e), we would expect that if a longer imbibition time (more than two months) was given, more *h3.3ko* seeds may germinate. Therefore, there seems to be phenotypic variations among these *h3.3ko* seeds. These 25% of *h3.3ko* seeds could be merely those happen to germinate within the two-month period we gave, but otherwise, they may not be significantly different from other *h3.3ko* seeds. Nevertheless, we have added ATAC-seq analysis of imbibed *h3.3ko* seeds, the results showed that the chromatin accessibility at the 5' gene end in the imbibed *h3.3ko* remained much lower than that in WT (Figure 6c and Supplemental Figure S12b). Considering that the H3.3 accumulation levels at these regions were gradually reduced during imbibition (Figure 6a), this further supports that H3.3 in mature/maturing seeds are critical for the establishment of chromatin accessibility.

Additional comments:

There are numerous grammatical errors which impair clear understanding of

what is written. A number of these are addressed below.

Response: We thank the reviewer for many value suggestions. We have tried our best to improve our writing.

The use of “pioneer” as a descriptive term seems inappropriate. “Pioneer” transcription factors are described as such because they have the ability to alter gene expression in the context of otherwise refractory chromatin. In this usage, chromatin is the template rather than the actor. Conferring “actor” status to H3.3 with the designation of “pioneer” impedes conceptual clarity. The authors are instead advised to use a distinct nomenclature that captures the foundational role of chromatin as the substrate upon which other factors (transcription factors and remodelers) act.

Response: We thank the reviewer for this suggestion. We have removed “pioneer” from the manuscript.

Line 39: The meaning of the following sentence is unclear as written: “At physiological level, embryos entering seed maturation stages have gradually obtained their germination capacity.”

Response: We appreciate the reviewer for pointing out this issue. We have changed this sentence to “Embryo obtains its germination capacity during the late seed maturation stage”.

Line 63: please provide specific plant reference for histone H3 phylogeny.

Response: We have included a reference showing amino acid sequence comparison between *Arabidopsis* H3.1 and H3.3 (Line 76).

Line 158: Use of “necessary” is preferred here rather than indispensable. Otherwise, this statement presumes the inability to identify suppressor mutations (for example), for which there is no evidence.

Response: We thank the reviewer for this suggestion. We have changed “indispensable” to “necessary”.

Line 161: Please define first use of “mature” for isolation of seeds here and again in Materials and Methods so that these experiments can be replicated.

Response: We thank the reviewer for this suggestion. We have defined mature seeds in the text (Line 170-173), and also in the Materials and Methods section.

Line 194: Please give number of histone genes, types, and some measure of extent of overrepresentation in place of “many histone genes”.

Response: We have removed this data as advised by other reviewers.

Line 203: Suggest “underrepresented” or “depleted” in place of “deprived”

Response: We thank the reviewer for this suggestion. We have changed “deprived” to “depleted”.

Line 209: Please specify using own RNA-seq data here and in Sup. Fig.3b

Response: We have removed these results in the revised manuscript, as they may not directly connect with the major focus of this study.

Line 268 and 271: Is Figure 5e supposed to be cited twice?

Response: We have removed the previous Figure 5e and instead added more typical loci to show increased transcript levels around their 3' gene end (Figure 5f).

Line 273: The authors appear to begin addressing cryptic transcription here without directly addressing this point. Doing so here (in the context of posing a testable hypothesis that is not about antisense transcripts) would help to make this transition a bit easier for the reader.

Response: We thank the reviewer for this suggestion. We have added a hypothesis here (Line 366-368).

Line 318 and following: It is unclear why the authors focus on the loss of the H3.3 peak at the 5' end and do not address the dramatic increase in H3.3 enrichment in gene bodies during germination in agreement with reference 14 from animals and the general concept of a replacement histone. The data certainly seem to warrant this observation.

Response: We thank the reviewer for raising this question. We focused on the 5' enriched H3.3 in mature seeds because 1) based on RNA-seq data, the transcriptomes between WT and *h3.3ko* were already drastically different at the mature seed stage (Figure 2b and 2e). 2) We have performed additional experiments to show that expressing H3.3 during the late seed maturation stage could rescue the germination defects of *h3.3ko*, but inducing H3.3 at the germination stage only could not (Figure 3a-3f). Since H3.3 is not essential for seed formation, these results suggest that H3.3 likely becomes critical from the late seed maturation stage, and thus we focused on the mature seed stage (and the 5' enrichment of H3.3 at this stage) because this is the earliest time point when we could distinguish *h3.3ko* from WT. This is not to say that H3.3 deposited in the gene body during germination is not important, rather the major focus of this study is to investigate the essential function of H3.3 at the earliest possible stage.

Line 329: Suggest replacing “is completely depleted from” with “enrichment is not detected in”

Response: We thank the reviewer for this suggestion. We have modified our writing here.

Line 358: “before the”

Response: Revised into “before the”.

Line 379: “this possibly involves”

Response: Revised into “this possibly involves”.

Line 393: “that may facilitate”

Response: We have modified the writing for this part.

Line 463: please define “mature” (e.g. age and treatment of seeds).

Response: We have defined “mature seeds” in both “Results” and “Materials and Methods” sections.

Line 468: Missing reference to Supp. Table 2 here to determine number of mapped reads.

Response: We used SAMtools to determine the numbers of mapped reads, the reference has been added.

Line 485: Please provide citation that demonstrates that freezing tissue and then treating with formaldehyde gives equivalent results to treating with formaldehyde and then freezing as is typically done. Reference 71 may state that same thing was done here, but would like to see citation with data that demonstrates that the order of these steps is interchangeable.

Response: We performed fixation immediately after grounding with liquid nitrogen because we consider that formaldehyde may not be able to penetrate into especially mature seeds in a short period of time (typically 10 minutes for fixation), while a longer fixation time may cause over-fixation. Other studies (Zhao et al., doi:10.1093/plphys/kiab224, Wollmann et al., doi:10.1371/journal.pgen.1002658) also used this way for H3.3 ChIP-seq analysis in vegetative tissues, and the results are similar to that obtained by doing fixation first (Stroud et al., doi: 10.1073/pnas.1203145109). In addition, H2A.Z ChIP-seq profiles obtained by doing this way (Xue et al., 10.1016/j.molp.2021.07.001, Yelagandula et al., doi:10.1016/j.cell.2014.06.006) are similar to other published H2A.Z ChIP-seq results (e.g. Coleman-Derr and Zilberman, doi:10.1371/journal.pgen.1002988). We have included these references in the “Materials and Methods/ChIP-seq” section.

Line 498: Number of mapped reads in Supp. Table 2 are substantially below ENCODE guidelines. Please explicitly address in manuscript.

Response: We thank the reviewer for raising this point. For the current encode standards, the histone experiments require 20 or 45 million reads (narrow-peak or broad-peak) for the human and mouse genomes. The human genome size is ~3.4Gb, which is about 25 times bigger than the *Arabidopsis* genome (~130Mb). Therefore, our reads have already provided enough coverage for the *Arabidopsis* genome. Please also see examples for mapped reads in other *Arabidopsis* ChIP-seq studies (Bieluszewski et al., doi: 10.1038/s41467-021-27882-5, Yin et al., doi: 10.1038/s41467-020-20614-1).

Line 512: How did authors “chop” mature seeds for isolation of nuclei? Were they hydrated before this? If yes, for how long? The chromatin is likely changing during this time.

Response: We thank the reviewer for raising this question. Matures seeds (dry seeds) were put into lysis buffer for only 5 minutes before chopping. Imbibed seeds and seedlings were chopped immediately after putting into the buffer.

Line 526: Number of mapped reads in Supp. Table 2 are substantially below ENCODE guidelines. Please explicitly address in manuscript.

Response: As mentioned above, the *Arabidopsis* genome is substantially smaller than that of human or mouse. The reads we obtained have already provided enough coverage for the *Arabidopsis* genome.

Line 886: “Expressing” should be used in place of “expression”

Response: Revised to “expressing”.

References: 48 and 53 are repeats

Response: We have made corrections in the manuscript.

Figure 1: Is there a citation providing data that the HTR13 promoter used is functional? Otherwise, the absence of an impact is easily interpreted as a consequence of using a non-functional promoter.

Response: We thank the reviewer for raising this point. The same *HTR13* promoter was successfully used for the complementation of the *h3.1kd* mutant (Jiang and Berger, doi: 10.1126/science.aan4965). We have now provided this reference (Line 142-144).

Figure 2b: Please specify if cited data are from embryo proper or entire seed.

Response: We have specified in the figure legend that the data was generated with embryos.

Figure 2d and 2e: Please confirm that inset bar refers to log₂ fold change.

Response: We have removed these figures based on other reviewers' suggestions. In a similar figure (Figure 6d), we have specified that the normalization is based on Row Z-score.

Figure 4c: Please specify how different regions are defined in Materials and Methods.

Response: Promoter refers to accessible regions located less than 2kb upstream of transcription start sites (TSS), downstream refers to accessible regions located less than 1kb downstream of transcription end sites (TES), Intergenic region refers to accessible regions located more than 2kb upstream of TSS and more than 1kb downstream of TES. We have now added their definitions in the Materials and Methods section.

Figure 5e: Please specify what type of gene (up or down, 5' or 3') CTR1 is supposed to exemplify.

Response: We have removed CTR1 and instead added more typical loci to show increased chromatin accessibility and transcript levels around their 3' gene end (Figure 5f).

Figure 6: Key needs to be provided for designation of types of samples.

Response: We have added the designation of these samples in the legend of Figure 2a.

Figure 6d: Please specify what type of gene (up or down, 5' or 3') phyB is supposed to exemplify.

Response: We have removed this loci based on other reviewers' suggestions.

Figure 6e: Please confirm that these are wild-type genes. Does this include both 5' and 3'? Genes that go up and down? Unclear what readers should conclude from these data as presented.

Response: We have now included the expression profiles of these genes during germination in both WT and *h3.3ko* (Figure 6d). These are genes associated with accessibility decreased regions in mature *h3.3ko* seeds compared with WT. These results suggest that H3.3-established chromatin accessibility at these genes is likely required for their transcriptional control during germination.

Figure 7: As noted above, no data are presented that H2A.Z is necessary for open chromatin or that TFs do not bind in H3.3ko seedlings. In the absence of such data, this model seems speculative.

Response: We thank the reviewer for raising this point. As mentioned above, we have added the ATAC-seq results in the *h2a.z* mutant, and moved the TF binding motif assay to the discussion section. In Figure 7, we have used the dashed border for TFs to indicate this possibility. We have also added the possibility that besides H2A.Z, 5' and 3' specific histone modifications may also contribute to the H3.3-mediated chromatin accessibility control.

Reviewers' Comments:

Reviewer #1:

Remarks to the Author:

The authors have addressed most of my concerns. I have following suggestions for improvement of the manuscript:

1. In Supplemental Figure 1d, all the genes driven by the HTR13 promoter were expressed at very low levels, it is likely that the HTR13 promoter is not active at the mature seed stage. The authors should confirm the HTR13 promoter works well in the transgenic plants in this study. The transcription levels of HTR13 driven genes at the vegetative stages should be tested to confirm this.
2. In the supplemental text, the "Figure 5" should be replaced by "Supplemental Figure 5".

Reviewer #2:

Remarks to the Author:

This revised version reads much better than the original one. It is clear that authors have made a significant effort to improve the text. They have included new results, in particular:

- heatmaps,
- the material used in each experiment,
- new data points for Fig. 2,
- analysis of Pol II, among others.

Therefore, I consider that authors have addressed satisfactorily all the suggestions made to the original submission.

Reviewer #1 (Remarks to the Author):

The authors have addressed most of my concerns. I have following suggestions for improvement of the manuscript:

Response: We thank the reviewer for all the very helpful suggestions and comments.

1. In Supplemental Figure 1d, all the genes driven by the HTR13 promoter were expressed at very low levels, it is likely that the HTR13 promoter is not active at the mature seed stage. The authors should confirm the HTR13 promoter works well in the transgenic plants in this study. The transcription levels of HTR13 driven genes at the vegetative stages should be tested to confirm this.

Response: We thank the reviewer for this suggestion. We have now performed gene expression analysis in vegetative tissues (7-day-old seedlings). The results show that the *HTR13* promoter could successfully drive gene expression in our transgenic lines (Supplemental Figure S1e). This further confirms that the *HTR13* promoter activity is very low in mature seeds (Figure 3d and Supplemental Figure S1d).

2. In the supplemental text, the “Figure 5” should be replaced by “Supplemental Figure 5”.

Response: We thank the reviewer for pointing out this mistake, we have changed the text accordingly.

Reviewer #2 (Remarks to the Author):

This revised version reads much better than the original one. It is clear that authors have made a significant effort to improve the text. They have included new results, in particular:

- heatmaps,
- the material used in each experiment,
- new data points for Fig. 2,
- analysis of Pol II, among others.

Therefore, I consider that authors have addressed satisfactorily all the suggestions made to the original submission.

Response: We thank the reviewer for all the constructive comments that helped to improve this manuscript.

Summary of additional comments on the response to Reviewer #3:

Reviewer #1 did not think that sufficient data was provided to show correlation between 5' H3.3, chromatin accessibility and transcriptional regulation during germination. Reviewer #1 commented that transcription factor binding to

promoters of genes essential for germination and post-embryonic development (e.g. ABA and auxin-related genes) had not been compared in wild-type and mutants lacking H3.3. The reviewer suggested that if antibodies against transcription factors cannot be obtained, the use of tagged transcription factors would be helpful to confirm the correlation between transcript levels and chromatin accessibility. Reviewer #1 also commented that there was insufficient evidence to show that H3.3 loading at 5' and 3' gene ends results in different gene regulation.

Response: We thank the reviewer for these comments. The TF binding assay suggested requires the generation of tagged TF lines in WT Col first and then crossed into *h3.3ko* to ensure the same transgenic line is used for comparison in Col and *h3.3ko*, which will take a long time to perform. We agree that this experiment would help to confirm the reduced chromatin accessibility and gene misregulation in *h3.3ko*. However, it may not provide further insights into how H3.3 controls chromatin accessibility, and we think that this point is beyond the scope of revising the present study, which focuses on the essential requirement of H3.3 in post-embryonic development and chromatin accessibility.

We understand the reviewer's other concerns and we have further modified the text to weaken the causal connections between changes in chromatin accessibility, gene transcription and post-embryonic development in *h3.3ko*. We also acknowledge that H3.3 may directly or indirectly regulate chromatin accessibility and the exact role of H3.3 in chromatin regulation and post-embryonic development remains to be investigated.

Reviewer #2 noted that while the majority of the concerns of reviewer #3 had been sufficiently addressed, the manuscript did make strong claims based on correlation and it would be advisable to tone down and make clear where evidence was correlative.

Response: We thank the reviewer for this suggestion. We have toned down the text and made indications where evidence was correlative.

Reviewers' Comments:

Reviewer #1:

Remarks to the Author:

The authors have addressed my concerns correctly.